# Follow-the-Perturbed-Leader for Decoupled Bandits: Best-of-Both-Worlds and Practicality

Chaiwon Kim [* 1]    Jongyeong Lee [* 2]    Min-hwan Oh [1]

## Abstract

We study the decoupled multi-armed bandit problem, where the learner separately selects one arm for exploration and one, possibly different, arm for exploitation at each round. In this setting, the loss of the explored arm is observed but not incurred, whereas the loss of the exploited arm is incurred without being observed. We propose an efficient Follow-the-Perturbed-Leader (FTPL) policy that achieves Best-of-Both-Worlds (BOBW) guarantee with constant regret in the stochastic regime and optimal $\mathcal{O}(\sqrt{KT})$ regret in the adversarial regime. A key feature of our method is that it completely avoids both the convex optimization required by prior BOBW policies and the resampling procedures typically used in FTPL bandit policies. This allows FTPL to fully realize its computational efficiency advantages, leading to substantial reductions in computational cost. We empirically confirm that our policy not only improves the runtime but also demonstrates superior regret performance in both regimes.

## 1. Introduction

The multi-armed bandit (MAB) problem (Thompson, 1933; Robbins, 1952; Lai & Robbins, 1985) is a fundamental framework for sequential decision-making, with applications in areas such as recommendation systems (Brodén et al., 2017; Zhou et al., 2017), communication systems (Maghsudi & Hossain, 2016; Li et al., 2020), and dynamic pricing (Misra et al., 2019). Consider the standard MAB problem, in which a learner selects one of $K$ arms at each round over a time horizon $T$. The learner aims to mini-

---
[*]Equal contribution  [1]Seoul National University, Seoul, Korea [2]Korea Institute of Science and Technology, Seoul, Korea. Correspondence to: Chaiwon Kim <snukcw128@snu.ac.kr>, Jongyeong Lee <jongyeong@kist.re.kr>, Min-hwan Oh <minoh@snu.ac.kr>.

*Proceedings of the 43rd International Conference on Machine Learning*, Seoul, South Korea. PMLR 306, 2026. Copyright 2026 by the author(s).

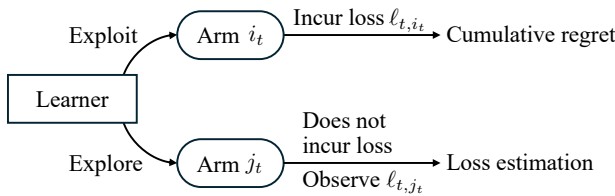

*Figure 1.* An illustration of decoupled multi-armed bandits, where exploitation and exploration are *decoupled* in each round.

mize cumulative regret, defined as the difference between the total loss incurred by the learner and that incurred by the best fixed arm in hindsight. At each round $t \in \{1, ..., T\}$, only the loss $\ell_{t,i_t}$ of the selected arm $i_t \in \{1, ..., K\}$ is observed and incurred. Thus, the learner must balance selecting arms that currently appear to have low expected loss (exploitation) with sampling arms whose losses remain uncertain, even if they currently appear suboptimal (exploration). In other words, each decision must simultaneously serve both exploitation and exploration, so the two objectives are *coupled* within each round.

However, the standard MAB model does not capture scenarios in which exploration can be performed separately from exploitation. For instance, in ultra-wideband communication systems, the learner can sense a channel different from the one used for transmission in order to observe feedback while avoiding interference with the ongoing transmission, which motivated the study of decoupled bandits by Avner et al. (2012). Another example arises when a real-time system operates alongside a high-fidelity simulator: the learner can explore in the simulator while exploiting in the real system, as in sim-to-real transfer for robotics (Zhao et al., 2020), without degrading real-world performance. A related example appears in recommendation systems (Che et al., 2025), where, given user contexts (e.g., preferences), a platform can explore on a random subset of users to update its policy while exploiting for the remaining users by serving the best-known items.

To model such scenarios, Avner et al. (2012) introduced the *decoupled* bandit setting (see Figure 1), where the learner can select two arms at each round: one for incurring the loss without observing it, and one for observing the loss without incurring it. This decoupling of exploration and ex-

ploitation recovers the standard multi-armed bandits, when the learner is restricted to select the same arm for both objectives. Note that this framework differs from pure explorations, in which the primary focus is on exploration to identify the best (good) arm (Even-Dar et al., 2006; Jourdan et al., 2023). It is also different from explore-then-commit style policies, which divide exploration and exploitation into distinct phases, performing only one of the two at each round (Garivier et al., 2016).

In the adversarial decoupled bandits, Avner et al. (2012) established a lower bound of $\Omega(\sqrt{KT})$, matching that of the standard multi-armed bandits (Auer et al., 2002). This result indicates that the two problems are similarly challenging in the adversarial regime. Against an oblivious adversary, their Exp3-type policy obtained a near-optimal adversarial regret of $\mathcal{O}(\sqrt{KT \ln K})$. In the stochastic setting with a unique optimal arm, they achieved a regret of $\mathcal{O}(\sqrt{T \ln K})$. However, this result is highly suboptimal, as even an anytime sampling rule designed for pure exploration tasks attains a time-independent cumulative regret of $\tilde{\mathcal{O}}(K^3/\Delta_{\min}^2)$ in the same setting, despite being aimed at minimizing the expected simple regret (Jourdan et al., 2023). Here, $\Delta_{\min} = \min_{i:\Delta_i>0} \Delta_i$ denotes the minimum suboptimality gap, where $\Delta_i = \mathbb{E}[\ell_{\cdot,i}] - \min_j \mathbb{E}[\ell_{\cdot,j}]$.

In addition to these limitations, the update rules for arm-selection probabilities and choice of learning rates proposed by Avner et al. (2012) require prior knowledge of both the time horizon and the environment. In practice, however, the nature of the environment is typically unknown, which motivates the design of policies that guarantee (near-)optimal performance across all possible environments, known as the Best-of-Both-Worlds (BOBW) guarantee (Bubeck & Slivkins, 2012).

Tsallis-INF, based on the Follow-the-Regularized-Leader (FTRL) framework, is a prominent BOBW policy for the multi-armed bandits (Zimmert & Seldin, 2021). Based on this policy, Rouyer & Seldin (2020) proposed Decoupled-Tsallis-INF, which also achieves BOBW guarantees in the decoupled bandit settings: optimal $\mathcal{O}(\sqrt{KT})$ regret in the adversarial regime and near-optimal time-independent regret of $\mathcal{O}(K/\Delta_{\min})$ in the stochastic regime. This result shows a significant improvement over Avner et al. (2012) and also outperforms the optimal bound for the standard stochastic bandits, which scales logarithmically with the horizon $T$ as $\mathcal{O}(\sum_{i:\Delta_i>0} \log T/\Delta_i)$.

Despite its strong theoretical guarantees, a practical drawback of the FTRL framework is the need to solve a convex optimization problem at every round to compute arm-selection probabilities, which can be computationally intensive. This has motivated interest in more computationally efficient alternatives, such as the Follow-the-Perturbed-Leader (FTPL) framework that selects an arm by adding random perturbations instead of solving optimizations, of which Exp3 is a special case (Auer et al., 2002; Bubeck & Cesa-Bianchi, 2012). In standard multi-armed bandits, recent work has shown that FTPL can achieve BOBW guarantees without requiring any convex optimization (Honda et al., 2023; Lee et al., 2024). In decoupled bandits, however, BOBW guarantees are obtained only by FTRL policy that requires solving optimizations (Rouyer & Seldin, 2020), while FTPL-type policy are known to achieve only suboptimal regret bounds (Avner et al., 2012). Hence, the following research question arise:

*Can we achieve BOBW guarantees for decoupled bandits while improving computational efficiency, without sacrificing regret performance in both regimes?*

**Contributions.** A key computational challenge of FTRL and FTPL policies in bandit problems lies in computing the arm-selection probability vector $w$, which is required both (i) to select an arm and (ii) to construct an unbiased loss estimator, since an importance-weighted (IW) loss estimator is typically used. In FTRL, these probabilities are obtained by solving a convex optimization problem at each round. By contrast, FTPL avoids convex optimization by selecting arms via random perturbations rather than explicitly computing probabilities. This is sufficient for (i), but creates a difficulty for (ii), since the IW estimator requires the probability of the selected arm. To estimate this, FTPL policies usually employ geometric resampling (Neu & Bartók, 2016) or its efficient variant (Chen et al., 2025), which incur a per-step average cost of $\mathcal{O}(K^2)$ or $\mathcal{O}(K \log K)$, respectively.

However, such resampling methods cannot be directly extended to the decoupled setting, where exploration is performed separately from exploitation. To the best of our knowledge, all existing decoupled bandit policies set the exploration probability as a function of the exploitation probability. Hence, directly employing this approach to FTPL policies poses a technical obstacle, since $w$ of FTPL is not available. In particular, if one directly adapt these designs to FTPL, e.g., by estimating the full vector via resampling, it would incur at least $\mathcal{O}(K^2 \log K)$ per-step cost, thereby eliminating the computational benefits of FTPL and potentially making it less efficient than FTRL.

In this paper, we introduce an alternative way to design the exploration probabilities for decoupled bandits that does not require explicit value of the arm-selection (exploitation) probability vector and can be computed using only currently available estimates. Based on this idea, we propose a decoupled FTPL policy that achieves BOBW *without convex optimization or resampling*, attaining the same regret order as Decoupled-Tsallis-INF while reducing computational cost. Our main contributions are as follows:

- We introduce a new FTPL-based policy that achieves

BOBW guarantees without convex optimization or re-sampling, where we attain $\mathcal{O}(\sqrt{KT})$ regret in the adversarial regime (Theorem 3.1) and $\mathcal{O}(K/\Delta_{\min})$ regret in the stochastic regime (Theorem 3.2).

- We show that the proposed policy achieves superior empirical performance even with faster runtime compared to existing BOBW policy.

Beyond these contributions, our approach to designing exploration probabilities can be of independent interest. In particular, it enables FTPL to fully realize its computational efficiency advantages by completely avoiding the use of re-sampling algorithms. Moreover, our analysis suggests that *surrogate probabilities can be used in place of exact probability vectors* to obtain the same order of regret guarantees. Consequently, these ideas can be applicable to other algorithmic frameworks, such as the Prod family (Cesa-Bianchi et al., 2007; Zimmert & Marinov, 2024), not restricted to FTPL, enabling comparable regret performance with improved computational efficiency.

## 2. Preliminaries

**Notation.** Let $T \in \mathbb{N}$ and $K \in \mathbb{N}$ denote the time horizon and the number of arms, respectively. For $n \in \mathbb{N}$, we use the shorthand $[n] := \{1, \ldots, n\}$. Let $\mathbf{0}$ and $\mathbf{1}$ denote the all-zeros vector and all-ones vector in $\mathbb{R}^K$ and $e_i$ denote the $i$-th standard basis vector in $\mathbb{R}^K$. For an event $A$, we write $\mathbb{1}[A]$ to denote its indicator function, which equals $1$ if $A$ occurs and $0$ otherwise. We also use the notation $x \wedge y := \min\{x, y\}$.

### 2.1. Problem setting

In the decoupled bandits, the learner selects an arm $i_t \in [K]$ to exploit, and an arm $j_t \in [K]$ to explore at each round $t \in [T]$, which may be the same or different. Then, the learner suffers $\ell_{t,i_t}$ without observing it, and observes $\ell_{t,j_t}$ without suffering it. Let $w_{t,i}$ be the exploitation probability and $p_{t,i}$ denote the exploration probability of arm $i$ at round $t$. The performance of a policy is measured by the pseudo-regret, defined as

$$\text{Reg}(T) = \mathbb{E}\left[\sum_{t=1}^{T} \ell_{t,i_t}\right] - \min_{i \in [K]} \mathbb{E}\left[\sum_{t=1}^{T} \ell_{t,i}\right]$$
$$= \mathbb{E}\left[\sum_{t=1}^{T} \langle \ell_t, w_t - e_{i^*} \rangle\right], \quad (1)$$

where $i^* = \arg\min_{i \in [K]} \mathbb{E}[\sum_{t=1}^{T} \ell_{t,i}]$ denotes the optimal arm in hindsight, assumed to be unique. The expectation $\mathbb{E}[\cdot]$ is taken over the randomness of policy and environment.

In this paper, we consider two environments for generating loss vectors $\ell_t = (\ell_{t,1}, \ldots, \ell_{t,K}) \in [0,1]^K$ at each round:

the adversarial regime (Auer et al., 2002) and stochastically constrained adversarial (SCA) regime (Wei & Luo, 2018). In the adversarial regime, loss vectors are determined by an adaptive adversary, in response to the learner's past actions. In the SCA regime, the environment may adjust the parameters of the arms (e.g., means) over rounds in response to the learner's past actions $\{i_s\}_{s=1}^{t-1}$. However, it is constrained to maintain fixed differences in the expected losses between any pair of arms, i.e., $\mathbb{E}[\ell_{t,i} - \ell_{t,j}] = \Delta_{i,j}$ for all $i, j, t$. Let $\Delta_i = \Delta_{i,i^*}$ denote the suboptimality gap of arm $i$, where $i^* = \arg\min_i \Delta_{i,1}$ holds in this regime. Therefore, this regime includes the pure stochastic setting as a special case, and its pseudo-regret satisfies

$$\text{Reg}(T) = \mathbb{E}\left[\sum_{t=1}^{T} \sum_{i \neq i^*} \Delta_i w_{t,i}\right]. \quad (2)$$

Since only the partial feedback is available, the learner constructs an unbiased estimator $\hat{\ell}_t$ of the loss vector, typically using an importance-weighted (IW) estimator, based on the observed feedback $\ell_{t,j_t}$ of the explored arm. Specifically, the IW loss estimator is defined as $\hat{\ell}_{t,i} = \ell_{t,i}\mathbb{1}[j_t = i]p_{t,i}^{-1}$ and then $\hat{L}_{t,i} = \sum_{s=1}^{t-1} \hat{\ell}_{s,i}$ denotes the estimated cumulative loss up to round $t-1$.

### 2.2. Previous approaches in decoupled bandits

Avner et al. (2012) proposed a decoupled bandit policy that uses an exploitation strategy based on Exp3 (Auer et al., 2002) and an exploration strategy designed to minimize the variance of the loss estimates, which scales as $\sum_i w_{t,i}/p_{t,i}$:

$$w_{t,i} = \frac{(1-\gamma)g_{t,i}}{\sum_{j=1}^{K} g_{t,j}} + \frac{\gamma}{K} \text{ and } p_{t,i} = \frac{\sqrt{w_{t,i}}}{\sum_{j=1}^{K} \sqrt{w_{t,i}}}, \quad (3)$$

where $\gamma$ is a parameter that depends on the learning rate $\eta$ and the number of arms $K$. While this policy can be applied to both regimes, the update rule for $g_{t,i}$ and the choice of learning rate $\eta$ differ across regimes, implying that the policy requires prior knowledge of the environment.

For BOBW guarantees, Rouyer & Seldin (2020) adopted $\beta$-Tsallis-INF policy with $\beta \in (0,1)$ as the exploitation strategy, a well-known BOBW policy in standard multi-armed bandits (Zimmert & Seldin, 2021). For exploration, they employed a strategy similar to that of Avner et al. (2012). Together, these form the Decoupled-Tsallis-INF policy:

$$w_t = \arg\min_{w \in \mathcal{S}_K} \left\{ \langle w, \hat{L}_t \rangle - \frac{1}{\eta_t} \sum_{i \in [K]} \frac{w_i^{\beta} - \beta w_i}{\beta(1-\beta)} \right\},$$
$$p_t = \left[ \frac{w_{t,i}^{1-\beta/2}}{\sum_{j \in [K]} w_{t,j}^{1-\beta/2}} \right]_{i \in [K]}. \quad (4)$$

Here, $\mathcal{S}_K = \{w \in [0,1]^K : \|w\|_1 = 1\}$ denotes the $(K-$

1)-dimensional probability simplex and the learning rate is $\eta_t = \mathcal{O}(t^{-1/2})$.

This policy achieves BOBW guarantees, with $\mathcal{O}(\sqrt{KT})$ adversarial regret for $\beta \in (0, 1)$ and time-independent regret $\mathcal{O}(K/\Delta_{\min})$ for $\beta \in (0, 2/3]$ in the SCA regime, significantly improving over previous Exp3-type policy. By construction, $\beta$-Tsallis-INF converges to Exp3 as $\beta \to 1$[1] and $p_t$ coincides with that of Avner et al. (2012) when $\beta = 1$. In this sense, Decoupled-Tsallis-INF roughly recovers (3) by tuning $\beta$. However, computing $w_t$ in (4) involves solving a convex optimization step, increasing the computational cost as the price for improved regret guarantees.

*Remark* 2.1. Recently, Jin et al. (2023) proposed an FTRL policy with a hybrid regularizer that combines a log-barrier and Tsallis entropy, together with arm-dependent learning rates, achieving an improved regret bound of $\mathcal{O}(\sqrt{\sum_i K/\Delta_i^2})$ in the SCA regime. While this approach is theoretically appealing, solving optimizations with such hybrid regularizers, especially those involving a log-barrier regularizer, incurs substantially heavier computational costs, as it requires solving a convex optimization at each round. In contrast, Tsallis-INF admits a solution that can be efficiently solved via Newton's method.

Recall that decoupled bandits were originally motivated by communication systems, where sensing and transmission decisions are often required to be made on very short time scales, making per-step computational efficiency important. However, our experiment in Appendix E.2 shows that directly using convex optimization algorithms incur significantly higher runtime, around 130 times slower even in $K = 2$, than our proposed policy. These results highlight the practical advantages of our approach, despite achieving near-optimal regret guarantees.

## 3. FTPL for decoupled bandits

In this section, we elaborate on technical challenges that arise in applying FTPL to decoupled bandits and present our method to overcome them.

### 3.1. Technical challenges

A common feature of previous decoupled bandit policies is that the exploration probability $p_{t,i}$ is computed using the exploitation probability $w_{t,i}$. In FTPL, however, $w_{t,i}$ generally lacks a closed-from expression, except in special cases such as FTPL with Gumbel perturbations, where the induced exploitation probability coincides with the multinomial logit model, i.e., the Exp3 policy. Although using Exp3 for exploitation is convenient, it results in suboptimal

performance in the stochastic regime (Avner et al., 2012). Instead, to obtain BOBW guarantees with FTPL, it is natural to adopt a Fréchet-type perturbation, due to its correspondence with $\beta$-Tsallis-INF (Kim & Tewari, 2019; Lee et al., 2025), an exploitation strategy known to achieve BOBW guarantees in multi-armed bandits, even though $w_{t,i}$ does not have a closed form (Honda et al., 2023; Lee et al., 2024; Chen et al., 2025).

A natural idea is to estimate $w_{t,i}$ via geometric resampling (GR), used in several bandit settings to construct the IW estimator (Neu & Bartók, 2016; Chen et al., 2025). In GR, after an arm $i_t$ is selected, the learner repeatedly resamples the perturbations until the same $i_t$ is selected again with those resampled perturbations, where the number of resampling steps becomes an unbiased estimator of $1/w_{t,i_t}$.

However, GR is designed to estimate only the probability of the selected arm $i_t$, rather than the full vector $w_t$. This makes direct application of GR with previous exploration policy infeasible, since computing $p_t$ requires estimates $w_{t,i}$ for all arms. Even if one could recover $w_t$ via repeated resampling, the average computational cost would increase by a factor of $K$, yielding a per-step cost of at least $\mathcal{O}(K^3)$ or $\mathcal{O}(K^2 \log K)$, depending on the method. Moreover, for arms with very small $w_{t,i}$, the required number of resampling iterations becomes large, as it scales as $1/w_{t,i}$.

### 3.2. Proposed policy

To overcome these challenges, we propose an alternative exploration policy for FTPL, which avoids both resampling steps and convex optimization steps and thus achieves considerable computational improvement. The key idea is to replace the arm-selection probability vector $w_t$ with a surrogate quantity that can be computed only from the currently available estimates.

**Exploitation.** At each round $t \in [T]$, the learner selects an arm $i_t$ to exploit according to the FTPL policy:

$$
\begin{aligned}
i_t &= \arg\min_{i \in [K]} \left\{ \hat{L}_{t,i} - \frac{r_{t,i}}{\eta_t} \right\} \\
&= \arg\min_{i \in [K]} \left\{ \underline{\hat{L}}_{t,i} - \frac{r_{t,i}}{\eta_t} \right\}, \; r_{t,i} \sim \mathcal{P}_\alpha, \forall i \in [K] \quad (5)
\end{aligned}
$$

where $\underline{\hat{L}}_t = \hat{L}_t - \mathbf{1} \cdot \min_{i \in [K]} \hat{L}_{t,i} \in [0, \infty)^K$ represents the loss-gap vector and $\eta_t$ denotes the learning rate specified later. Here, $r_t = (r_{t,1}, \ldots, r_{t,K})$ is a random perturbation vector whose components are sampled i.i.d. from the Pareto distribution $\mathcal{P}_\alpha$ with shape parameter $\alpha > 1$. The density function $f$ and distribution function $F$ of $\mathcal{P}_\alpha$ are given as

$$
f(x) = \frac{\alpha}{x^{\alpha+1}}, \quad F(x) = 1 - \frac{1}{x^\alpha}, \quad x \in [1, \infty),
$$

---

[1]Strictly speaking, it coincides with the version of Exp3 in Bubeck & Cesa-Bianchi (2012), whereas the original version (Auer et al., 2002) includes an additional $\gamma/K$ term.

---

**Algorithm 1** FTPL for decoupled bandits

---

1: **Input:** Shape $\alpha > 1$ and rule to decide learning rate $\eta_t$.
2: **Initialization:** $\hat{L}_1 = \mathbf{0}$.
3: **for** $t = 1$ **to** $T$ **do**
4:     Sample $(r_{t,1}, \ldots, r_{t,K})$ i.i.d. from $\mathcal{P}_\alpha$.
5:     Select $i_t$ as in (5) and incur $\ell_{t,i_t}$.
6:     Explore $j_t \sim p_t$ in (7) and observe $\ell_{t,j_t}$.
7:     Update $\hat{L}_{t+1} = \hat{L}_t + \ell_{t,j_t} p_{t,j_t}^{-1} e_{j_t}$.
8: **end for**

---

respectively. Then, the exploitation probability of arm $i \in [K]$ given $\hat{L}_t$ can be expressed by $w_{t,i} = \phi_i(\eta_t \hat{L}_t)$, where

$$\phi_i(\eta_t \hat{L}_t) := \mathbb{P}_{r_t \sim \mathcal{P}_\alpha^K}\left[ i = \arg\min_{i \in [K]} \left\{ \hat{\underline{L}}_{t,i} - \frac{r_{t,i}}{\eta_t} \right\} \right]$$

$$= \int_1^\infty f(z + \eta_t \hat{\underline{L}}_{t,i}) \prod_{j \neq i} F(z + \eta_t \hat{\underline{L}}_{t,j}) \mathrm{d}z, \quad (6)$$

which cannot be expressed in the closed-form.

**Exploration.** In addition to FTPL exploitation, the learner selects an arm $j_t$ for exploration according to the probability distribution $p_t$, defined as

$$p_{t,i} = \frac{q_{t,i}}{\sum_{j \in [K]} q_{t,j}}, \quad \text{where}$$

$$q_{t,i} = \left( \frac{1}{1 + \eta_t \hat{\underline{L}}_{t,i}} \wedge \frac{1}{\sigma_{t,i}^{1/\alpha}} \right)^{\frac{\alpha+1}{2}}, \quad (7)$$

and $\sigma_{t,i}$ denotes the rank of $\hat{L}_{t,i}$ among $\{\hat{L}_{t,j}\}_{j \in [K]}$, with $\sigma_{t,i} = 1$ for the smallest and $K$ for the largest value (ties are broken arbitrarily). It is obvious that $p_t$ is computable directly from $\hat{L}_t$ and $\eta_t$ without additional procedures, with at most $O(K \log K)$ per-step cost due to sorting $\{\hat{L}_{t,j}\}_j$.

Given the correspondence between the $\beta$-Tsallis entropy and Fréchet-type perturbations with $\alpha = 1/(1 - \beta)$, $q_{t,i}$ can be viewed as an approximation of $w_{t,i}^{1/2+1/(2\alpha)}$, which roughly corresponds to $w_{t,i}^{1-\beta/2}$ in (4). Our approach of approximating $w_{t,i}$ using a tight upper bound (see Lemma D.2 in Appendix for details) may be of independent interest for efficiently approximating arm-selection probabilities of FTPL beyond the decoupled setting. The pseudo-code of the overall procedure is given in Algorithm 1.

### 3.3. Theoretical guarantees

The following results establish the regret guarantees and show BOBW guarantees, with the first theorem showing that Algorithm 1 is optimal in the adversarial regime.

**Theorem 3.1.** *In the adversarial regime, Algorithm 1 with $\alpha > 1$ and $\eta_t = cK^{\frac{1}{\alpha} - \frac{1}{2}}/\sqrt{t}$ for $c > 0$ satisfies*

$$\mathrm{Reg}(T) \leq \mathcal{O}(\sqrt{KT}).$$

The proof of Theorem 3.1 is given in Appendix B. This result matches the lower bound of Avner et al. (2012) up to constants, and is therefore order optimal. In the next theorem, we analyze the regret of Algorithm 1 in the SCA regime including the stochastic setting. Note that Theorem 3.1 requires only the bounded loss assumption, i.e., $\ell_t \in [0,1]^K$ and does not require a constant gap assumption used in the SCA regime, which encompasses the classical stochastic regime as a special case.

**Theorem 3.2.** *In the stochastically constrained adversarial regime with a unique best arm $i^*$, Algorithm 1 with $\alpha \in (1,3]$ and $\eta_t = cK^{\frac{1}{\alpha} - \frac{1}{2}}/\sqrt{t}$ for $c > 0$ satisfies*

$$\mathrm{Reg}(T) \leq \mathcal{O}\left( \left( \sum_{t=1}^T \sum_{i \neq i^*} \frac{\sqrt{K}\Delta_i^{1-\alpha}}{t^{\frac{\alpha}{2}}} \right) + \frac{K}{\Delta_{\min}} \right). \quad (8)$$

A proof sketch of Theorem 3.2 is provided in Section 3.4, with the detailed proof in Appendix C. The theorem focuses on $\alpha \in (1,3]$, since for $\alpha > 3$ the dependence on $K$ worsens from $\sqrt{K}$ to $K^{\frac{\alpha-2}{\alpha-1}}$. Such degradation, which also arises in the BOBW FTRL policy with $\beta \in (2/3, 1)$, is undesirable (Rouyer & Seldin, 2020). Note that the bound in (8) becomes independent of the time horizon $T$ for $\alpha \in (2,3]$, since $\sum_{t=1}^T t^{-\alpha/2}$ converges as $T \to \infty$ whenever $\alpha > 2$. In particular, $\alpha = 3$ minimizes this bound, achieving near-optimal regret as follows.

**Corollary 3.3.** *In the same setting as in Theorem 3.2, Algorithm 1 with $\alpha = 3$ and $\eta_t = cK^{-\frac{1}{6}}/\sqrt{t}$ for $c > 0$ satisfies*

$$\mathrm{Reg}(T) \leq \mathcal{O}\left( \sqrt{\frac{K}{\Delta_{\min}} \sum_{i \neq i^*} \frac{1}{\Delta_i}} + \frac{K}{\Delta_{\min}} \right).$$

In general, our bound coincides with Rouyer & Seldin (2020) for $\beta = 2/3$. Given the correspondence $\alpha = 1/(1 - \beta)$ noted earlier, this similarity is natural. However, under specific conditions on the suboptimality gaps, their overall bound can become tighter, as their first term reduces from $\mathcal{O}(K/\Delta_{\min})$ to $\mathcal{O}\left(\sqrt{\frac{K}{\Delta_{\min}} \sum_{i \neq i^*} \frac{1}{\Delta_i}}\right)$. In contrast, our first term already attains their best possible result without any extra assumptions. The relative looseness of our result comes from the analysis of the second term, which scales as $K/\Delta_{\min}$, whereas FTRL achieves the sharper $\sqrt{K}/\Delta_{\min}$ rate.

We expect that the improved regret for FTPL-based policy in the SCA regime can be achieved by introducing arm-dependent learning rates, as in FTRL-based methods (Jin et al., 2023). However, this would require significantly intricate analysis, since arm-dependent learning rates in FTPL

are indeed equivalent to using arm-dependent perturbations with arm-independent learning rates in (5), leading to FTPL with non-i.i.d. perturbations. We leave the development of such an analysis for future work, as it is beyond the scope of this paper.

### 3.4. Proof sketch of the regret in the SCA regime

Here, we provide a proof sketch of Theorem 3.2. We begin by decomposing the pseudo-regret in (1), which can be seen as a reduction of Lemmas 3.3 and 3.4 in Zhan et al. (2025) from the semi-bandit setting to the multi-armed bandit setting. Compared to prior analyses in multi-armed bandits (Honda et al., 2023; Lee et al., 2024; 2025), our decomposition avoids an additional $\log T$ factor. For completeness, the detailed proof is given in Appendix A.1.

**Lemma 3.4** (Regret decomposition). *Let $\{\eta_t\}_{t \in [T]}$ be a sequence of positive, decreasing learning rates and $\eta_0 = \infty$. Then, Algorithm 1 with $\alpha > 1$ satisfies*

$$
\mathrm{Reg}(T) \le \sum_{t=1}^{T} \mathbb{E}\Big[\Big\langle \hat{\ell}_t, \phi(\eta_t \hat{L}_t) - \phi(\eta_t \hat{L}_{t+1}) \Big\rangle \Big]
$$
$$
+ \sum_{t=1}^{T+1} \left( \frac{1}{\eta_t} - \frac{1}{\eta_{t-1}} \right) \mathbb{E}_{r_t \sim \mathcal{P}_\alpha^K}[r_{t,i_t} - r_{t,i^*}]. \quad (9)
$$

Following the convention, we refer to the first and second term of (9) as the stability term and penalty term, respectively. The penalty term can be bounded from above as follows, providing a slightly tighter bound than Lee et al. (2024). The proof is provided in Appendix A.3.

**Lemma 3.5.** *For any $t \in [T]$, Algorithm 1 with $\alpha > 1$ satisfies*

$$
\mathbb{E}_{r_t \sim \mathcal{P}_\alpha^K}\Big[r_{t,i_t} - r_{t,i^*}\Big|\hat{L}_t\Big]
$$
$$
\le \frac{\alpha}{\alpha - 1} \sum_{i \ne i^*} \frac{1}{(1 + \eta_t \underline{\hat{L}}_{t,i})^{\alpha - 1}} \wedge C_\alpha K^{\frac{1}{\alpha}},
$$

*where $C_\alpha = \frac{2\alpha^3 + (e-2)\alpha^2}{(\alpha-1)(2\alpha-1)}$.*

The stability term can be bounded from above as follows.

**Lemma 3.6.** *For any $t \in [T]$, $i \in [K]$ and $q_t$ defined in (7), Algorithm 1 with $\alpha > 1$ satisfies*

$$
\mathbb{E}\Big[\hat{\ell}_{t,i}\Big(\phi_i(\eta_t \hat{L}_t) - \phi_i(\eta_t \hat{L}_{t+1})\Big)\Big|\hat{L}_t\Big]
$$
$$
\le e(\alpha + 1)\eta_t \sum_{j \in [K]} q_{t,j} q_{t,i}.
$$

We provide the proof of Lemma 3.6 in Appendix A.2. While the proof structure follows the previous analysis (Lee et al., 2024), our construction of $p_{t,i}$ allows the stability term to

be upper bounded in terms of $q_{t,i}$, which enables BOBW guarantees in the decoupled setting.

Under the learning rate in Theorems 3.1 and 3.2, the order of the bound on penalty term is never larger than that of the stability for $\alpha \in (1, 3]$, making the stability term dominant in the regret. This observation is consistent with Rouyer & Seldin (2020), who analyzed the case $\beta \in (0, 2/3]$.

As discussed in Corollary 3.3, one may consider the use of arm-dependent (Jin et al., 2023) or stability–penalty matching (Tsuchiya et al., 2023; Ito et al., 2024) learning rates, instead of the current simple learning rate, to equalize contributions of the two terms. We expect this could yield (possibly improved) BOBW guarantees of FTPL for general $\alpha > 1$. However, to the best of our knowledge, such designs have not been explored for FTPL, in contrast to FTRL frameworks, which leverage the explicit probability vector $w_t$, an approach that cannot be directly extended to FTPL due to its inherent structure. Therefore, developing a principled counterpart of such adaptive learning rate designs for FTPL is an important direction for future work, and we believe our approach of replacing $w_t$ with the efficiently computable $q_t$ provides a promising direction for such extensions.

**Proof sketch.** In the SCA regime, the regret is written in terms of $\Delta_i$ and exploitation probability $w_{t,i}$ as in (2). Therefore, to derive a desired bound, we need to express the stability and penalty terms in terms of $w_{t,i}$ and $\Delta_i$. Using analysis techniques similar to those in previous FTPL analyses for the standard multi-armed bandits (Honda et al., 2023; Lee et al., 2024), we define the following events:

$$
D_t := \left\{ \sum_{i \ne i^*} \frac{1}{(2^{1/\alpha} + \eta_t \underline{\hat{L}}_{t,i})^\alpha} \le \frac{1}{2} \right\}. \quad (10)
$$

This event is introduced only for the sake of analysis, and Algorithm 1 does not need to know whether $D_t$ occurs. On $D_t$, it implies that $\underline{\hat{L}}_{t,i^*} = 0$, indicating that the optimal arm $i^*$ has been accurately identified based on the information so far, and we can bound $q_{t,i}$ in terms of $w_{t,i}$ as follows.

$$
q_{t,i} \le \left( \frac{1}{1 + \eta_t \underline{\hat{L}}_{t,i}} \right)^{\frac{\alpha+1}{2}} \lesssim w_{t,i}^{\frac{1}{2} + \frac{1}{2\alpha}} \lesssim w_{t,i}^{1 - \frac{1}{\alpha}}, \forall i \ne i^*,
$$

where the second step follows from Lemma D.2 and the last step holds when $\alpha \in (1, 3)$. This relationship is a key technical lemma in our analysis and also provides the intuition behind our choice of $q_t$. In particular, the design of $D_t$ enables us to establish an explicit relationship between $q_{t,i}$ and $w_{t,i}$, which may be of independent interest.

In Appendix D, we show that the stability term of the optimal arm $i^*$ is bounded by that of the suboptimal arms on $D_t$, i.e., $\mathcal{O}(\eta_t \sum_{j \in [K]} q_{t,j} \cdot \sum_{i \ne i^*} q_{t,i})$. Therefore, by

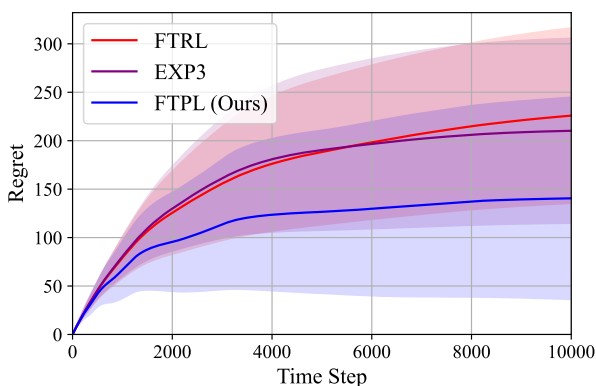

*Figure 2.* Adversarial regret with $\Delta = 0.125$

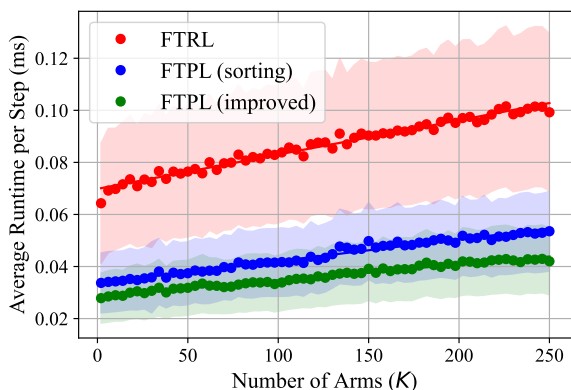

*Figure 3.* Computation time per-step (ms), 100 runs

Lemma 3.6, we have

$\text{Reg}(T)$

$$\leq \mathbb{E}\left[\sum_{t=1}^{T}\mathcal{O}\left(\mathbb{1}[D_t]\frac{K^{\frac{1}{\alpha}-\frac{1}{2}}}{\sqrt{t}}\sum_{j\in[K]}q_{t,j}\sum_{i\neq i^*}w_{t,i}^{1-\frac{1}{\alpha}}\right)\right]$$

$$+\mathbb{E}\left[\sum_{t=1}^{T}\mathcal{O}\left(\mathbb{1}[D_t^c]\sqrt{\frac{K}{t}}\right)\right]$$

$$\leq \mathbb{E}\left[\sum_{t=1}^{T}\mathcal{O}\left(\mathbb{1}[D_t]\frac{K^{\frac{1}{2\alpha}}}{\sqrt{t}}\sum_{i\neq i^*}w_{t,i}^{1-\frac{1}{\alpha}}+\mathbb{1}[D_t^c]\sqrt{\frac{K}{t}}\right)\right],$$

where the last step follows from the definition of $q_{t,i}$, which implies $\sum_i q_{t,i} \leq \sum_i i^{-\frac{1}{2}-\frac{1}{2\alpha}} \leq \frac{2\alpha}{\alpha-1}K^{\frac{1}{2}-\frac{1}{2\alpha}}$. On the complement event $D_t^c$, we apply the uniform bounds from Lemmas 3.5 and 3.6, which leads to terms depending only on $K$, $t$, and $\alpha$. By the definition of pseudo-regret in (2), we obtain

$$\text{Reg}(T) \geq \mathbb{E}\left[\sum_{t=1}^{T}\mathbb{1}[D_t]\sum_{i\neq i^*}\Delta_i w_{t,i}+\mathbb{1}[D_t^c]0.196\Delta_{\min}\right],$$

where the value of 0.196 is introduced by Lemma D.3, which provides the lower bound on $w_{t,i_t}$. By applying the self-bounding technique, we have

$$\text{Reg}(T) \leq \mathbb{E}\left[\sum_{t=1}^{T}\mathcal{O}\left(\sum_{i\neq i^*}\left(\frac{K^{\frac{1}{2\alpha}}}{\sqrt{t}}w_{t,i}^{1-\frac{1}{\alpha}}-\Delta_i w_{t,i}\right)\right)\right]$$

$$+\mathbb{E}\left[\sum_{t=1}^{T}\mathcal{O}\left(\sqrt{\frac{K}{t}}-\Delta_{\min}\right)\right].$$

Since $\max_{w\in[0,1]}\frac{Aw^{1-\frac{1}{\alpha}}}{\sqrt{t}}-\Delta_i w \leq \mathcal{O}(A^\alpha \Delta_i^{1-\alpha}t^{-\alpha/2})$, the regret satisfies

$$\text{Reg}(T) \leq \mathcal{O}\left(\left(\sum_{t=1}^{T}\sum_{i\neq i^*}\sqrt{K}\Delta_i^{1-\alpha}t^{-\frac{\alpha}{2}}\right)+\frac{K}{\Delta_{\min}}\right),$$

which concludes the proof.

## 4. Numerical experiments

We evaluate the empirical performance and computational efficiency of our policy, Algorithm 1, under two regimes in the decoupled setting: adversarial and stochastic regimes. All experiments are conducted for 1000 independent repetitions unless otherwise specified, with the time horizon $T = 10000$, $\alpha = 3$ for Algorithm 1, and $\beta = 2/3$ for Decoupled-Tsallis-INF (Rouyer & Seldin, 2020). We also provide the results of FTRL with Shannon entropy, which corresponds to the case of $\beta = 1$ for Decoupled-Tsallis-INF. This policy is closely related to the decoupled Exp3 policy considered in Avner et al. (2012) as discussed in Section 2.2, where arm-selection probabilities is written in the closed form. For simplicity, we denote Algorithm 1 by FTPL, $\beta = 2/3$ and $\beta = 1$ instances of Decoupled-Tsallis-INF by FTRL and EXP3, respectively. Although the constant $c$ in the learning rate $\eta_t$ can be tuned either to minimize regret bound analytically or empirically, we set $c = 2$ following prior studies (Zimmert & Seldin, 2021; Honda et al., 2023; Lee et al., 2024). The shaded regions in the figures indicate one standard deviation. Details on the implementation and the additional results including real-world data-based simulation are given in Appendix E.3.

### 4.1. Implementation of FTPL

The computation of $p_t$ in (7) requires the rank $\sigma_{t,i}$ of $\hat{L}_{t,i}$. As discussed in Section 3.2, a straightforward implementation recomputes these ranks at every round by sorting the values $\{\hat{L}_{t,i}\}_i$, which incurs an average complexity of $\mathcal{O}(K\log K)$. However, this is unnecessarily inefficient since the loss estimator is updated only for the selected arm $i_t$, and therefore the ranks of all other arms can change by at most one.

Consequently, instead of re-sorting all arms, it suffices to locate the updated rank $\sigma_{t+1,i_t}$ of the played arm while maintaining the previous rank in memory $\Theta(K)$. The optimized version locates the new rank of played arm via binary search in $\mathcal{O}(\log K)$ and then updates the rank of affected block of arms, which takes $\mathcal{O}(K)$ in the worst case.

### 4.2. Adversarial regime

For the adversarial regime, we follow the setup of Zimmert & Seldin (2021). Specifically, the losses are generated from Bernoulli distributions, where mean losses of the optimal arm and all suboptimal arms alternate between $(0, \Delta)$ and $(1 - \Delta, 1)$, with the duration of each phase growing exponentially as $\lfloor 1.6^n \rfloor$, where $n$ denotes the phase index. We consider an eight-armed bandit with a unique optimal arm under $\Delta = 0.125$. To further demonstrate its scalability and robustness, we extend this evaluation to 16 distinct instances by varying both $K$ and $\Delta$, with the results provided in Appendix E.2.

The first experiment (Figures 2 and 3) evaluates the performance of BOBW policies under $\Delta = 0.125$. Specifically, Figure 2 presents the empirical regret performance, showing that our policy achieves lower cumulative regret.

Figure 3 shows the computational efficiency of FTPL relative to FTRL. Here, following the implementation details in Section 4.1, we refer to the straightforward sorting-based implementation as FTPL(sorting), and to the optimized implementation as FTPL(improved). The average per-step runtime is measured as the number of arms increases, with $K \in \{4x + 2 : x = 0, 1, \ldots, 63\}$, and we additionally report a linear regression fit, as a small number of outliers appear for certain values of $K$, likely due to computational resource variability.

While the FTRL policies generally require solving a convex optimization problem, in the Tsallis entropy case, $w_t$ can be computed efficiently using Newton's method with warm start (Zimmert & Seldin, 2021). Even with these efficient implementations, Figure 3 shows that the runtime of FTRL is slower than that of FTPL as also reflected by the larger slope of the line even when compared with the sorting-based implementation of FTPL, which has $\mathcal{O}(K \log K)$ average complexity.

Although the optimization can be solved by iterative Newton's method that costs $\mathcal{O}(K)$ per iteration, as noted by Chen et al. (2025), the required number of iterations to satisfy its regret guarantee has not been formally characterized, which makes the total computational cost of FTRL remain theoretically unclear. While the difference between $\mathcal{O}(K)$ and $\mathcal{O}(K \log K)$ may not be clearly visible in moderate $K$, the key observation is that the two implementations of FTPL have fundamentally different theoretical complexities. In

this context, FTRL appears to show a similar or slightly less favorable trend than the $\mathcal{O}(K \log K)$ implementation due to the iterative nature of Newton's method.

To further demonstrate the efficiency of Newton's method, we evaluate the per-step runtime of FTRL using a splitting conic solver (SCS) for $K \in \{2^i : i \in [6]\}$ in Appendix E.2 (O'Donoghue et al., 2016). In this setting, we observe that the runtime increases by at least 100 times.

### 4.3. Stochastic regime

In this setting, we consider the two five-armed bandit instances with Bernoulli rewards and a unique optimal arm. The first is an easy instance with $\mu_1 = [0.55, 0.6, 0.45, 0.3, 0.2]$ and $\Delta_{\min} = 0.05$. The second one is a difficult instance with $\mu_2 = [0.905, 0.91, 0.89, 0.908, 0.88]$, which we construct to have smaller gaps $\Delta_{\min} = 0.002$ between arms while all arms have high mean rewards, making it harder to distinguish the optimal arm.

In Figure 4, the regret of all policies gradually converges, which is consistent with the theoretical analysis. However, as the problem instances become more challenging from Figure 4 to Figure 5, it seems more rounds are required for convergence in the harder setting. Nevertheless, in Figures 2, 4, and 5, all BOBW policies show the intended robustness across both stochastic and adversarial regimes. Among them, FTPL achieves the best empirical performance while also offering improved computational efficiency.

## 5. Conclusion

We proposed a practically efficient FTPL policy with Pareto perturbations that achieves BOBW guarantees in the decoupled multi-armed bandit problems. Our policy achieves optimal regret $\mathcal{O}(\sqrt{KT})$ in the adversarial regime, and a near-optimal time-independent regret bound of $\mathcal{O}(K/\Delta_{\min})$ in the stochastically constrained adversarial regime. Our result improves upon the optimal regret bound of the standard stochastic multi-armed bandits, $\mathcal{O}(\sum_{i \neq i^*} \log T/\Delta_i)$, which is time-dependent.

In addition to these theoretical strengths, we avoid both the convex optimization step in previous BOBW policy, and the resampling step typically required in FTPL for estimating arm-selection probabilities. As a result, our policy shows improved computational efficiency compared to Decoupled-Tsallis-INF, while achieving better empirical performance across both regimes.

A key idea of our policy is the replacement of $w_t$ with an efficiently computable approximation $q_t$. We expect that this approach can be used to design the refined learning rates (e.g., adaptive learning rates) for FTPL: by using an approximation of $w_t$, one can adjust the learning rate in a

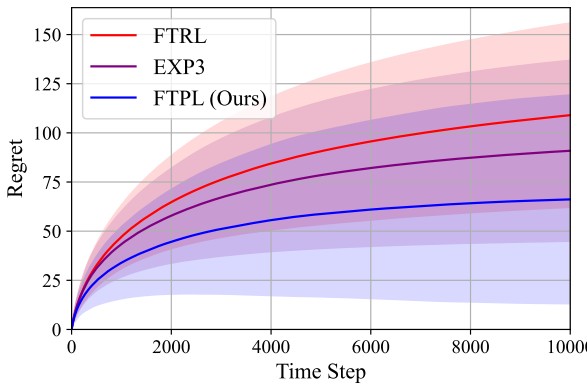

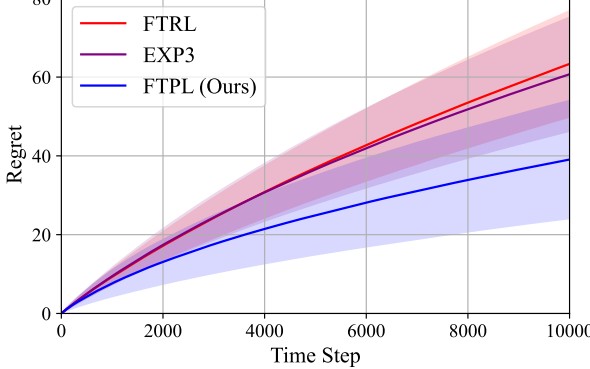

*Figure 4.* Stochastic regret with $\mu_1 = [0.55, 0.6, 0.45, 0.3, 0.2]$ and $\Delta_{\min} = 0.05$

*Figure 5.* Stochastic regret with $\mu_2 = [0.905, 0.91, 0.89, 0.908, 0.88]$ and $\Delta_{\min} = 0.002$

way analogous to FTRL frameworks, where the learning rate is explicitly determined by $w_t$. This perspective potentially serves as a foundation for establishing BOBW guarantees for FTPL beyond the MAB setting.

## Acknowledgements

CK and JL were supported by the National Research Foundation of Korea (NRF) grant funded by Korea government (MSIT) (No. RS-2024-00395303). JL was also supported by the grant Nos. 2024-00460980; and 2025-02304717 (IITP) funded by the Korea government (the Ministry of Science and ICT). MO was supported by the NRF and the IITP (Nos. RS-2022-NR071853, RS-2023-00222663, RS-2025-25463302).

## Impact Statement

This paper focuses on theoretical aspects of bandit problems, and we believe there are no direct negative ethical or societal impacts that should be highlighted here. Our research may facilitate the real-world applications of the FTPL policies, such as in recommendation systems.

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

# A. Omitted Proofs for Lemmas

In this section, we provide the detailed proofs for lemmas omitted in the main paper.

## A.1. Proof for the regret decomposition (Lemma 3.4)

While the overall proof is a straightforward adaptation of the arguments in the semi-bandit setting, particularly Lemmas 3.3 and 3.4 from Zhan et al. (2025), to the MAB setting, we provide all details here for completeness.

We begin by recalling the connection between FTPL and FTRL, where it is known that FTPL can generally be expressed as FTRL with a specific corresponding regularizer (Abernethy et al., 2015; Suggala & Netrapalli, 2020). To formalize this, consider a convex potential function $\Phi : \mathbb{R}^K \to \mathbb{R}$ for $\phi$ defined as

$$\Phi(\lambda) = \mathbb{E}_r \left[ \max_{i \in [K]} \{\lambda_i + r_i\} \right],$$

so that the gradient of $\Phi$ satisfies $\nabla \Phi(\lambda) = \phi(-\lambda)$, where $\phi$ denotes the arm-selection probability function of FTPL. The convex conjugate (or Lagrange transform) of $\Phi$ is given by

$$\Phi^*(p) = \sup_{\lambda \in \mathbb{R}^K} \langle p, \lambda \rangle - \Phi(\lambda), \quad \text{for } p \in \text{Int}(\mathcal{P}_{K-1}),$$

where $\text{Int}(\mathcal{P}_{K-1})$ denotes the interior of the probability simplex of dimension $K - 1$. It is known that FTPL is equivalent to FTRL with regularizer $\Phi^*(p)$. By standard results in the convex analysis (see Zhan et al., 2025, Lemma G.1 and the references therein), one can see that $w_t = \nabla \Phi(-\eta_t \hat{L}_t)$ implies $-\eta_t \hat{L}_t \in \partial \Phi^*(w_t)$, and hence

$$w_t \in \arg\min_{x \in \mathcal{P}_{K-1}} \left\{ \Phi^*(x)/\eta_t + \left\langle x, \hat{L}_t \right\rangle \right\}.$$

It is worth noting that if $w_t$ lies on the boundary of the simplex, the gradient $\nabla \Phi^*(p)$ may not exist. Nevertheless, we consider minimization over $\mathcal{P}_{K-1}$ rather than $\text{Int}(\mathcal{P}_{K-1})$, since the regularizer $\Phi^*(p)$ remains well-defined even on boundary points, although its gradients may be unbounded.

**Lemma 3.4 (restated)** *Let $\{\eta_t\}_{t \in [T]}$ be a sequence of positive, decreasing learning rates and $\eta_0 = \infty$. Then, Algorithm 1 with $\alpha > 1$ satisfies*

$$\text{Reg}(T) \leq \sum_{t=1}^{T} \mathbb{E}\left[ \left\langle \hat{\ell}_t, \phi(\eta_t \hat{L}_t) - \phi(\eta_t \hat{L}_{t+1}) \right\rangle \right] + \sum_{t=1}^{T+1} \left( \frac{1}{\eta_t} - \frac{1}{\eta_{t-1}} \right) \mathbb{E}_{r_t \sim \mathcal{D}}[r_{t,i_t} - r_{t,i^*}].$$

*Proof.* Following the proof of Lemma 3.3 of Zhan et al. (2025), let $\Phi_t^*(x) = \Phi^*(x)/\eta_t + \left\langle x, \hat{L}_t \right\rangle$. By definition, we have $w_t \in \arg\min_{x \in \mathcal{P}_{K-1}} \Phi_t^*(x)$ and

$$\sum_{t=1}^{T} \left\langle w_t - e_{i^*}, \hat{\ell}_t \right\rangle$$

$$= \sum_{t=1}^{T} \left\langle w_t - w_{t+1}, \hat{\ell}_t \right\rangle + \sum_{t=1}^{T} \left\langle w_{t+1}, \hat{\ell}_t \right\rangle - \sum_{t=1}^{T} \left\langle e_{i^*}, \hat{\ell}_t \right\rangle$$

$$= \sum_{t=1}^{T} \left\langle w_t - w_{t+1}, \hat{\ell}_t \right\rangle + \sum_{t=1}^{T} \left( \Phi_{t+1}^*(w_{t+1}) - \frac{\Phi^*(w_{t+1})}{\eta_{t+1}} - \left( \Phi_t^*(w_{t+1}) - \frac{\Phi^*(w_{t+1})}{\eta_t} \right) \right)$$

$$\quad - \sum_{t=1}^{T} \left( \Phi_{t+1}^*(e_{i^*}) - \frac{\Phi^*(e_{i^*})}{\eta_{t+1}} - \left( \Phi_t^*(e_{i^*}) - \frac{\Phi^*(e_{i^*})}{\eta_t} \right) \right) \tag{11}$$

$$= \sum_{t=1}^{T} \left\langle w_t - w_{t+1}, \hat{\ell}_t \right\rangle + \sum_{t=1}^{T} (\Phi_t^*(w_t) - \Phi_t^*(w_{t+1})) + \sum_{t=2}^{T+1} \left( \frac{1}{\eta_t} - \frac{1}{\eta_{t-1}} \right) (\Phi^*(e_{i^*}) - \Phi^*(w_t))$$

$$\quad + \Phi_{T+1}^*(w_{T+1}) - \Phi_1^*(w_1) - \Phi_{T+1}^*(e_{i^*}) + \Phi_1^*(e_{i^*})$$

where (11) follows from the definition of $\hat{L}_t$ and $\Phi_t^*$ that

$$\left\langle w_{t+1}, \hat{\ell}_t \right\rangle = \left\langle w_{t+1}, \hat{L}_{t+1} - \hat{L}_t \right\rangle \quad \text{and} \quad \left\langle x, \hat{L}_t \right\rangle = \Phi_t^*(x) - \Phi^*(x)/\eta_t.$$

Since $\hat{L}_1 = \mathbf{0}$ and $\Phi_{T+1}^*(w_{T+1}) \leq \Phi_{T+1}^*(e_{i^*})$, we have

$$\sum_{t=1}^{T} \left\langle w_t - e_{i^*}, \hat{\ell}_t \right\rangle \leq \sum_{t=1}^{T} \left( \left\langle w_t - w_{t+1}, \hat{\ell}_t \right\rangle + \Phi_t^*(w_t) - \Phi_t^*(w_{t+1}) \right)$$
$$+ \sum_{t=2}^{T+1} \left( \frac{1}{\eta_t} - \frac{1}{\eta_{t-1}} \right) (\Phi^*(e_{i^*}) - \Phi^*(w_t)) + \frac{\Phi^*(e_{i^*}) - \Phi^*(w_1)}{\eta_1}.$$

For notational simplicity, let $\eta_0 = \infty$, which is not used in the policy and is introduced merely for the analysis. Then,

$$\sum_{t=1}^{T} \left\langle w_t - e_{i^*}, \hat{\ell}_t \right\rangle \leq \sum_{t=1}^{T} \left( \left\langle w_t - w_{t+1}, \hat{\ell}_t \right\rangle + \Phi_t^*(w_t) - \Phi_t^*(w_{t+1}) \right)$$
$$+ \sum_{t=1}^{T+1} \left( \frac{1}{\eta_t} - \frac{1}{\eta_{t-1}} \right) (\Phi^*(e_{i^*}) - \Phi^*(w_t)). \tag{12}$$

Let $D_\Phi$ denote the Bregman divergence associated with $\Phi$, which is defined by

$$D_\Phi(x, y) = \Phi(x) - \Phi(y) - \langle \nabla\Phi(y), x - y \rangle, \ \forall x, y \in \mathbb{R}^K.$$

Since $w_t \in \partial\Phi^*(-\eta_t \hat{L}_t)$ and $-\eta_t \hat{L}_t \in \partial\Phi(w_t)$, it holds

$$\Phi(-\eta_t \hat{L}_t) + \Phi^*(w_t) = \left\langle w_t, -\eta_t \hat{L}_t \right\rangle.$$

for all $t \in \mathbb{N}$. Then, we have

$$\Phi_t^*(w_t) - \Phi_t^*(w_{t+1}) = -\frac{1}{\eta_t} \left( \Phi^*(w_{t+1}) - \Phi^*(w_t) - \left\langle w_{t+1} - w_t, -\eta_t \hat{L}_t \right\rangle \right) \qquad \text{(by definition of } \Phi_t^*\text{)}$$
$$= -\frac{1}{\eta_t} \left( \left\langle w_{t+1}, -\eta_{t+1} \hat{L}_{t+1} \right\rangle - \Phi(-\eta_{t+1} \hat{L}_{t+1}) \right.$$
$$\left. + \left\langle w_t, \eta_t \hat{L}_t \right\rangle + \Phi(-\eta_t \hat{L}_t) - \left\langle w_{t+1} - w_t, -\eta_t \hat{L}_t \right\rangle \right)$$
$$= -\frac{1}{\eta_t} \left( \Phi(-\eta_t \hat{L}_t) - \Phi(-\eta_{t+1} \hat{L}_{t+1}) - \left\langle w_{t+1}, -\eta_t \hat{L}_t + \eta_{t+1} \hat{L}_{t+1} \right\rangle \right)$$
$$= -\frac{1}{\eta_t} D_\Phi(-\eta_t \hat{L}_t, -\eta_{t+1} \hat{L}_{t+1}). \qquad (\because w_{t+1} = \nabla\Phi(-\eta_{t+1} \hat{L}_{t+1}))$$

On the other hand, by definition, one can obtain (or see Lemma G.2 of Zhan et al. (2025))

$$D_\Phi(x, y) + D_\Phi(z, x) - D_\Phi(z, y) = \langle \nabla\Phi(x) - \nabla\Phi(y), x - z \rangle$$

for any $x, y, z \in \mathbb{R}^K$. Therefore, by letting $x = -\eta_t \hat{L}_{t+1}$, $y = -\eta_{t+1} \hat{L}_{t+1}$ and $z = -\eta_t \hat{L}_t$, we obtain

$$D_\Phi(-\eta_t \hat{L}_{t+1}, -\eta_{t+1} \hat{L}_{t+1}) + D_\Phi(-\eta_t \hat{L}_t, -\eta_t \hat{L}_{t+1}) - D_\Phi(-\eta_t \hat{L}_t, -\eta_{t+1} \hat{L}_{t+1})$$
$$= \left\langle \nabla\Phi(-\eta_t \hat{L}_{t+1}) - w_{t+1}, -\eta_t \hat{L}_{t+1} + \eta_t \hat{L}_t \right\rangle$$
$$= \left\langle \phi(-\eta_t \hat{L}_{t+1}) - w_{t+1}, -\eta_t \hat{\ell}_t \right\rangle,$$

which implies

$$
\begin{aligned}
&\left\langle w_t - w_{t+1}, \hat{\ell}_t \right\rangle + \Phi_t^*(w_t) - \Phi_t^*(w_{t+1}) \\
&= \frac{1}{\eta_t} \left\langle w_t - w_{t+1}, \eta_t \hat{\ell}_t \right\rangle - \frac{1}{\eta_t} D_\Phi(-\eta_t \hat{L}_t, -\eta_{t+1} \hat{L}_{t+1}) \\
&= \frac{1}{\eta_t} \left\langle w_t - \phi(-\eta_t \hat{L}_{t+1}) + \phi(-\eta_t \hat{L}_{t+1}) - w_{t+1}, \eta_t \hat{\ell}_t \right\rangle - \frac{1}{\eta_t} D_\Phi(-\eta_t \hat{L}_t, -\eta_{t+1} \hat{L}_{t+1}) \\
&= \left\langle w_t - \phi(-\eta_t \hat{L}_{t+1}), \hat{\ell}_t \right\rangle + \frac{1}{\eta_t} \left( \left\langle \phi(-\eta_t \hat{L}_{t+1}) - w_{t+1}, \eta_t \hat{\ell}_t \right\rangle - D_\Phi(-\eta_t \hat{L}_t, -\eta_{t+1} \hat{L}_{t+1}) \right) \\
&= \left\langle w_t - \phi(-\eta_t \hat{L}_{t+1}), \hat{\ell}_t \right\rangle - \frac{1}{\eta_t} \left( D_\Phi(-\eta_t \hat{L}_{t+1}, -\eta_{t+1} \hat{L}_{t+1}) + D_\Phi(-\eta_t \hat{L}_t, -\eta_t \hat{L}_{t+1}) \right) \\
&\leq \left\langle w_t - \phi(\eta_t \hat{L}_{t+1}), \hat{\ell}_t \right\rangle. \hspace{4cm} (\because D_\Phi(\cdot, \cdot) \geq 0)
\end{aligned}
$$

Since $w_t = \phi(\eta_t \hat{L}_t)$, it remains to control the second term in (12).

While it can be obtained by direct application of Lemma 3.4 of Zhan et al. (2025), we provide the corresponding proof here for completeness. By definition of $\Phi^*$ and $w_t = \nabla \Phi(-\eta_t \hat{L}_t)$, we have

$$
\begin{aligned}
\Phi^*(w_t) = -\left\langle \eta_t \hat{L}_t, w_t \right\rangle - \Phi(-\eta_t \hat{L}_t) &= -\mathbb{E}\left[ \left\langle \eta_t \hat{L}_t, e_{i_t} \right\rangle \right] + \mathbb{E}\left[ \min_i \eta_t \hat{L}_{t,i} - r_{t,i} \right] \\
&= -\mathbb{E}\left[ \left\langle \eta_t \hat{L}_t, e_{i_t} \right\rangle \right] + \mathbb{E}\left[ \left\langle \eta_t \hat{L}_t - r_t, e_{i_t} \right\rangle \right] \\
&= -\mathbb{E}[r_{t,i_t}].
\end{aligned}
$$

By definition of $\Phi$, we have $\Phi(\lambda) \geq \mathbb{E}_r[\langle r + \lambda, p \rangle]$ for any $p \in \mathcal{P}_{K-1}$ and $\lambda \in \mathbb{R}^K$. Hence,

$$
\begin{aligned}
\Phi^*(e_{i^*}) = \sup_{x \in \mathbb{R}^K} \langle x, e_i^* \rangle - \Phi(x) &\leq \sup_{x \in \mathbb{R}^K} \langle x, e_i^* \rangle - \mathbb{E}_r[\langle r + x, e_{i^*} \rangle] \\
&= -\mathbb{E}_r[\langle r, e_{i^*} \rangle] = -\mathbb{E}_r[r_{i^*}],
\end{aligned}
$$

which concludes the proof. $\hspace{10cm} \square$

### A.2. Proof for the stability term (Lemma 3.6)

**Lemma 3.6 (restated)** *For any $t \in [T]$, $i \in [K]$, Algorithm 1 with $\alpha > 1$ satisfies*

$$
\mathbb{E}\left[ \hat{\ell}_{t,i} \left( \phi_i(\eta_t \hat{L}_t) - \phi_i(\eta_t \hat{L}_{t+1}) \right) \Big| \hat{L}_t \right] \leq e(\alpha + 1)\eta_t \sum_{j \in [K]} q_{t,j} q_{t,i},
$$

*where $q_t$ is defined in (7).*

*Proof.* Let $\lambda \in \mathbb{R}^K$ and $\phi_i'(\lambda) = \frac{\partial \phi_i(\lambda)}{\partial \lambda_i}$, which is

$$
\begin{aligned}
\phi_i'(\lambda) &= \int_{-\min_j \lambda_j}^{\infty} -\frac{\alpha(\alpha+1)}{(z + \lambda_i + 1)^{\alpha+2}} \prod_{j \neq i} \left( 1 - \frac{1}{(z + \lambda_j + 1)^\alpha} \right) dz \\
&= \int_0^{\infty} -\frac{\alpha(\alpha+1)}{(z + \underline{\lambda}_i + 1)^{\alpha+2}} \prod_{j \neq i} \left( 1 - \frac{1}{(z + \underline{\lambda}_j + 1)^\alpha} \right) dz,
\end{aligned}
$$

where the underline denotes $\underline{\lambda} = \lambda - \mathbf{1} \cdot \min_j \lambda_j$. Note that $-\phi_i'(\lambda)$ is decreasing with respect to $\lambda_i$ and increasing with

respect to $\lambda_j$. Then, by definition, we have

$$
\mathbb{1}[i = j_t]\Big(\phi_i(\eta_t \hat{L}_t) - \phi_i(\eta_t \hat{L}_{t+1})\Big)
$$

$$
= \mathbb{1}[i = j_t] \int_0^{\eta_t \ell_{t,i} p_{t,i}^{-1}} -\phi_i'(\eta_t \hat{L}_t + x e_i) \mathrm{d}x
$$

$$
= \mathbb{1}[i = j_t] \int_0^{\eta_t \ell_{t,i} p_{t,i}^{-1}} -\phi_i'(\eta_t \hat{L}_t) \mathrm{d}x \qquad (\because \text{decreasing w.r.t. } \lambda_i)
$$

$$
= \mathbb{1}[i = j_t] \int_0^{\infty} \frac{\alpha(\alpha+1)\eta_t \ell_{t,i} p_{t,i}^{-1}}{(z + \underline{\lambda}_i + 1)^{\alpha+2}} \prod_{j \neq i}\left(1 - \frac{1}{(z + \underline{\lambda}_j + 1)^{\alpha}}\right)\mathrm{d}z.
$$

Let $I_{i,\alpha+2}(\lambda) = \int_0^{\infty} \frac{1}{(z+\lambda_i+1)^{\alpha+2}} \prod_{j \neq i}\left(1 - \frac{1}{(z+\lambda_j+1)^{\alpha}}\right)\mathrm{d}z$. Then,

$$
\mathbb{E}\Big[\hat{\ell}_{t,i}\Big(\phi_i(\eta_t \hat{L}_t) - \phi_i(\eta_t \hat{L}_{t+1})\Big)\Big|\hat{L}_t\Big] = \mathbb{E}\left[\frac{\ell_{t,i}\mathbb{1}[j_t = i]}{p_{t,i}}\Big(\phi_i(\eta_t \hat{L}_t) - \phi_i(\eta_t \hat{L}_{t+1})\Big)\Big|\hat{L}_t\right]
$$

$$
\leq \alpha(\alpha+1)\eta_t \mathbb{E}\left[\frac{\ell_{t,i}^2 \mathbb{1}[j_t = i]}{p_{t,i}^2} I_{i,\alpha+2}(\eta_t \underline{\hat{L}}_t)\Big|\hat{L}_t\right]
$$

$$
\leq \alpha(\alpha+1)\eta_t \mathbb{E}\left[\frac{I_{i,\alpha+2}(\eta_t \underline{\hat{L}}_t)}{p_{t,i}}\Big|\hat{L}_t\right],
$$

where the last inequality follows from $\ell_{t,i} \leq 1$ and $\mathbb{E}[\mathbb{1}[j_t = i]|\hat{L}_t] = p_{t,i}$.

By definition of $I_{i,\alpha+2}$, one can see that

$$
I_{i,\alpha+2}(\underline{\lambda}) \leq I_{i,\alpha+2}(\lambda^*), \quad \text{where} \quad \lambda_j^* = \begin{cases} \underline{\lambda}_i, & \sigma_j \leq \sigma_i, \\ \infty, & \sigma_j > \sigma_i, \end{cases}
$$

where $\sigma_i$ denotes the rank of $\lambda_i$ in the increasing order of $\lambda$, i.e., $\sigma_i < \sigma_j$ iff $\lambda_i \leq \lambda_j$ with arbitrary tie-breaking rule. In the later of the proof, we assume $\sigma_i = i$ without loss of generality for the simplicity, i.e., $\lambda_1 \leq \lambda_2 \leq \ldots, \lambda_K$. Then, we have

$$
I_{i,\alpha+2}(\lambda^*) = \int_0^{\infty} \frac{1}{(z + \underline{\lambda}_i + 1)^{\alpha+2}}\left(1 - \frac{1}{(z + \underline{\lambda}_i + 1)^{\alpha}}\right)^{i-1}\mathrm{d}z
$$

$$
= \frac{1}{\alpha}\int_0^{\frac{1}{(1+\underline{\lambda}_i)^{\alpha}}} w^{\frac{1}{\alpha}}(1 - w)^{i-1}\mathrm{d}w
$$

$$
= \frac{1}{\alpha}B\left(\frac{1}{(1+\underline{\lambda}_i)^{\alpha}}; 1 + \frac{1}{\alpha}, i\right),
$$

where $B(x; a, b) = \int_0^x t^{a-1}(1 - t)^{b-1}\mathrm{d}t$ denotes the incomplete Beta function. By elementary calculation, we obtain for any $x \in [0, 1]$

$$
B\left(x; 1 + \frac{1}{\alpha}, i\right) = \int_0^x t^{\frac{1}{\alpha}}(1 - t)^{i-1}\mathrm{d}t \leq \int_0^x t^{\frac{1}{\alpha}} e^{-t(i-1)}\mathrm{d}t
$$

$$
\leq e \int_0^x t^{\frac{1}{\alpha}} e^{-ti}\mathrm{d}t \qquad (\because x \in [0, 1])
$$

$$
= \frac{e}{i^{1+\frac{1}{\alpha}}}\gamma\left(1 + \frac{1}{\alpha}, xi\right),
$$

where $\gamma(a, x)$ denotes the lower incomplete gamma function. Since $\gamma(a, x) \leq \Gamma(a)$ for any $x > 0$, we have

$$
I_{i,\alpha+2}(\lambda^*) \leq \frac{e}{\alpha i^{1+\frac{1}{\alpha}}}\Gamma\left(1 + \frac{1}{\alpha}\right) \leq \frac{e}{\alpha i^{1+\frac{1}{\alpha}}}\Gamma(2) = \frac{e}{\alpha i^{1+\frac{1}{\alpha}}}. \tag{13}
$$

On the other hand, by Equation 8.10.2 of Olver et al. (2010), it holds that

$$\gamma\left(1 + \frac{1}{\alpha}, \frac{i}{(1 + \underline{\lambda}_i)^\alpha}\right) \leq \frac{\alpha}{\alpha + 1} \frac{i^{1/\alpha}}{(1 + \underline{\lambda}_i)} \min\left(1, \frac{i}{(1 + \underline{\lambda}_i)^\alpha}\right),$$

which implies

$$I_{i,\alpha+2}(\lambda^*) \leq \frac{e}{\alpha + 1} \frac{1}{(1 + \underline{\lambda}_i)i}. \tag{14}$$

Therefore, from (13) and (14), we obtain

$$\alpha(\alpha + 1)\mathbb{E}\left[\frac{I_{i,\alpha+2}(\eta_t \underline{\hat{L}}_t)}{p_{t,i}}\bigg| \hat{L}_t\right] \leq \frac{e(\alpha + 1)}{p_{t,i}i^{1+\frac{1}{\alpha}}} \wedge \frac{e\alpha}{p_{t,i}(1 + \eta_t \underline{\hat{L}}_{t,i})i}.$$

By definition of $p_t$ and $q_{t,i}$ in (7), when $\frac{1}{(1+\eta_t \underline{\hat{L}}_{t,i})} \leq \frac{1}{i^{1/\alpha}}$, we have

$$\frac{1}{p_{t,i}} \frac{1}{(1 + \eta_t \underline{\hat{L}}_{t,i})i} = \sum_j q_{t,j} \frac{\sqrt{i(1 + \eta_t \underline{\hat{L}}_{t,i})}}{(1 + \eta_t \underline{\hat{L}}_{t,i})i} = \sum_j q_{t,j} \frac{1}{\sqrt{i(1 + \eta_t \underline{\hat{L}}_{t,i})}} = \sum_j q_{t,j} q_{t,i}.$$

On the other hand, when $\frac{1}{(1+\eta_t \underline{\hat{L}}_{t,i})} \geq \frac{1}{i^{1/\alpha}}$, we have

$$\frac{1}{p_{t,i}i^{1+1/\alpha}} = \sum_j q_{t,j} \frac{\sqrt{i^{1+1/\alpha}}}{i^{1+1/\alpha}} = \sum_j q_{t,j} \frac{1}{\sqrt{i^{1+1/\alpha}}} = \sum_j q_{t,j} q_{t,i}.$$

Therefore, in any cases, we obtain

$$\mathbb{E}\left[\hat{\ell}_{t,i}\left(\phi_i(\eta_t \hat{L}_t) - \phi_i(\eta_t \hat{L}_{t+1})\right)\bigg| \hat{L}_t\right] \leq e(\alpha + 1)\eta_t \sum_{j \in [K]} q_{t,j} q_{t,i},$$

which concludes the proof. $\qquad\square$

### A.3. Proof for the penalty term (Lemma 3.5)

**Lemma 3.5 (restated)** *For any $t \in [T]$, Algorithm 1 with $\alpha > 1$ satisfies*

$$\mathbb{E}_{r_t \sim \mathcal{P}_\alpha^K}\left[r_{t,i_t} - r_{t,i^*}\bigg| \hat{L}_t\right] \leq \frac{\alpha}{\alpha - 1} \sum_{i \neq i^*} \frac{1}{(1 + \eta_t \underline{\hat{L}}_{t,i})^{\alpha-1}} \wedge C_\alpha K^{\frac{1}{\alpha}},$$

*where $C_\alpha = \frac{2\alpha^3 + (e-2)\alpha^2}{(\alpha-1)(2\alpha-1)}$.*

*Proof.* The proof of this lemma is almost the same as that of Lemma 12 of Lee et al. (2024), except that we provide a slightly tighter bound.

By the choice of Pareto perturbations, we have

$$\mathbb{E}\left[r_{t,i_t} - r_{t,i^*}\bigg| \hat{L}_t\right] \leq \sum_{i \neq i^*} \mathbb{E}\left[\mathbb{1}[I_t = i]r_{t,i}\bigg| \hat{L}_t\right]$$

$$= \int_1^\infty \sum_{i \neq i^*} \left(\frac{\alpha}{(z + \eta_t \underline{\hat{L}}_{t,i})^\alpha}\right) \prod_{j \neq i} \left(1 - \frac{1}{(z + \eta_t \underline{\hat{L}}_{t,j})^\alpha}\right) dz$$

$$\leq \int_1^\infty \sum_{i \neq i^*} \left(\frac{\alpha}{(z + \eta_t \underline{\hat{L}}_{t,i})^\alpha}\right) dz$$

$$= \frac{\alpha}{\alpha - 1} \sum_{i \neq i^*} \frac{1}{(1 + \eta_t \underline{\hat{L}}_{t,i})^{\alpha-1}}.$$

Let $k_\alpha(z) = \sum_{i \in [K]} \frac{1}{(z + \eta_t \hat{\underline{L}}_{t,i})^\alpha} \in (0, \frac{K}{z^\alpha}]$. Then,

$$
\int_1^\infty \sum_{i \neq i^*} \left( \frac{\alpha}{(z + \eta_t \hat{\underline{L}}_{t,i})^\alpha} \right) \prod_{j \neq i} \left( 1 - \frac{1}{(z + \eta_t \hat{\underline{L}}_{t,j})^\alpha} \right) \mathrm{d}z
$$

$$
\leq \int_1^\infty \sum_{i \neq i^*} \left( \frac{\alpha}{(z + \eta_t \hat{\underline{L}}_{t,i})^\alpha} \right) \exp \left( -\sum_{j \neq i} \left( 1 - \frac{1}{(z + \eta_t \hat{\underline{L}}_{t,j})^\alpha} \right) \right) \mathrm{d}z
$$

$$
\leq e\alpha \int_1^\infty \sum_{i \neq i^*} \left( \frac{\alpha}{(z + \eta_t \hat{\underline{L}}_{t,i})^\alpha} \right) e^{-k_\alpha(z)} \mathrm{d}z \leq e \int_1^\infty k_\alpha(z) e^{-k_\alpha(z)} \mathrm{d}z.
$$

Since $xe^{-x} \leq e^{-1}$ for $x \geq 0$ and $xe^{-x}$ is increasing for $x \leq 1$ and $k_\alpha(z)$ holds for $z \geq K^{1/\alpha}$, we have

$$
e\alpha \int_1^\infty k_\alpha(z) e^{-k_\alpha(z)} \mathrm{d}z \leq \alpha \int_1^{K^{1/\alpha}} 1 \mathrm{d}z + e \int_{K^{1/\alpha}}^\infty \alpha \frac{K}{z^\alpha} e^{-\frac{K}{z^\alpha}} \mathrm{d}z
$$

$$
= \alpha(K^{1/\alpha} - 1) + eK^{1/\alpha} \int_0^1 w^{-\frac{1}{\alpha}} e^{-w} \mathrm{d}w
$$

$$
= K^{1/\alpha} \left( \alpha + e\gamma \left( 1 - \frac{1}{\alpha}, 1 \right) \right) - \alpha,
$$

where $\gamma$ denotes the lower incomplete gamma function. By the same arguments in Lee et al. (2024, Appendix D.1.), it holds that

$$
\gamma \left( 1 - \frac{1}{\alpha}, 1 \right) \leq \frac{\alpha^2 (1 - e^{-1})}{(\alpha - 1)(2\alpha - 1)} + \frac{\alpha e^{-1}}{\alpha - 1} = \frac{(1 + e^{-1})\alpha^2 - e^{-1}\alpha}{(\alpha - 1)(2\alpha - 1)},
$$

which implies

$$
e\alpha \int_1^\infty k_\alpha(z) e^{-k_\alpha(z)} \mathrm{d}z \leq \left( \alpha + \frac{(e+1)\alpha^2 - \alpha}{(\alpha - 1)(2\alpha - 1)} \right) K^{\frac{1}{\alpha}} - \alpha
$$

$$
\leq \frac{2\alpha^3 + (e-2)\alpha^2}{(\alpha - 1)(2\alpha - 1)} K^{\frac{1}{\alpha}} - \alpha,
$$

which concludes the proof. $\qquad\square$

*Remark* A.1. While the additional $-\alpha$ term is not directly used in the analysis, it is easy to observe that the upper bound vanishes as $\alpha \to \infty$. This behavior is intuitive since larger values of $\alpha$ correspond to perturbation distributions with lighter right tails, increasingly concentrated around the left endpoint at 1. In the limit, as $\alpha \to \infty$, the perturbation converges to a Dirac delta function at 1, eliminating any randomness, i.e., the difference between perturbations becomes zero.

## B. Regret bound for adversarial bandits (Theorem 3.1)

**Theorem 3.1 (restated)** *In the adversarial regime, Algorithm 1 with $\alpha > 1$ and $\eta_t = cK^{\frac{1}{\alpha} - \frac{1}{2}}/\sqrt{t}$ for $c > 0$ satisfies* $\mathrm{Reg}(T) \leq \mathcal{O}(\sqrt{KT})$.

*Proof.* From Lemma 3.4, we have

$$
\mathrm{Reg}(T) \leq \sum_{t=1}^T \mathbb{E} \left[ \left\langle \hat{\ell}_t, \phi(\eta_t \hat{L}_t) - \phi(\eta_t \hat{L}_{t+1}) \right\rangle \right] + \sum_{t=1}^{T+1} \left( \frac{1}{\eta_t} - \frac{1}{\eta_{t-1}} \right) \mathbb{E}_{r_t \sim \mathcal{D}}[r_{t,i_t} - r_{t,i^*}].
$$

For the stability term (the first term), we have

$$\sum_{t=1}^{T} \mathbb{E}\Big[\Big\langle \hat{\ell}_t, \phi(\eta_t \hat{L}_t) - \phi(\eta_t \hat{L}_{t+1}) \Big\rangle\Big] = \mathbb{E}\left[\sum_{t=1}^{T} \sum_{i\in[K]} \mathbb{E}\Big[\hat{\ell}_{t,i}\Big(\phi_i(\eta_t \hat{L}_t) - \phi_i(\eta_t \hat{L}_{t+1})\Big)\Big|\hat{L}_t\Big]\right]$$

$$\leq \mathbb{E}\left[\sum_{t=1}^{T} e(\alpha+1)\eta_t \sum_{j\in[K]} q_{t,j} \sum_{i\in[K]} q_{t,i}\right] \qquad \text{(by Lemma 3.6)}$$

$$\leq \sum_{t=1}^{T} \frac{4\alpha^2(\alpha+1)e}{(\alpha-1)^2} \frac{cK^{\frac{1}{\alpha}-\frac{1}{2}}}{\sqrt{t}} K^{1-\frac{1}{\alpha}} \qquad (15)$$

$$= \sum_{t=1}^{T} \frac{4c\alpha^2(\alpha+1)e}{(\alpha-1)^2} \sqrt{\frac{K}{t}}$$

$$\leq \frac{8c\alpha^2(\alpha+1)e}{(\alpha-1)^2} \sqrt{KT}, \qquad (16)$$

where (15) follows by the definition of $q_{t,i}$,

$$\sum_{i\in[K]} q_{t,i} = \sum_{i\in[K]} \left(\frac{1}{1+\eta_t \hat{\underline{L}}_{t,i}} \wedge \frac{1}{\sigma_i^{1/\alpha}}\right)^{\frac{\alpha+1}{2}} \leq \sum_{i\in[K]} i^{-\frac{1}{2}-\frac{1}{2\alpha}} \leq \frac{2\alpha}{\alpha-1} K^{\frac{1}{2}-\frac{1}{2\alpha}}. \qquad (17)$$

For the penalty term (the second term), we have

$$\sum_{t=1}^{T+1} \left(\frac{1}{\eta_t} - \frac{1}{\eta_{t-1}}\right) \mathbb{E}_{r_t\sim\mathcal{P}_\alpha^K}[r_{t,i_t} - r_{t,i^*}]$$

$$= \frac{\mathbb{E}_{r_t\sim\mathcal{P}_\alpha^K}[r_{t,i_t} - r_{t,i^*}]}{\eta_1} + \sum_{t=2}^{T+1} \left(\frac{1}{\eta_t} - \frac{1}{\eta_{t-1}}\right) \mathbb{E}_{r_t\sim\mathcal{P}_\alpha^K}[r_{t,i_t} - r_{t,i^*}]$$

$$\leq \frac{\alpha\Gamma(1-1/\alpha)}{\alpha-1} \frac{\sqrt{K}}{c} + \sum_{t=2}^{T+1} \left(\frac{1}{\eta_t} - \frac{1}{\eta_{t-1}}\right) \mathbb{E}_{r_t\sim\mathcal{P}_\alpha^K}[r_{t,i_t} - r_{t,i^*}], \qquad (18)$$

where the inequality follows from Lemma 18 of Lee et al. (2024). For the second term in (18), we have

$$\sum_{t=2}^{T+1} \left(\frac{1}{\eta_t} - \frac{1}{\eta_{t-1}}\right) \mathbb{E}\Big[\mathbb{E}_{r_t\sim\mathcal{P}_\alpha^K}\Big[r_{t,i_t} - r_{t,i^*}\Big|\hat{L}_t\Big]\Big] \leq \frac{K^{\frac{1}{2}-\frac{1}{\alpha}}}{c} \sum_{t=2}^{T+1} \left(\sqrt{t} - \sqrt{t-1}\right) C_\alpha K^{\frac{1}{\alpha}}$$

$$= \frac{C_\alpha\sqrt{K}}{c} \left(\sqrt{T+1} - 1\right)$$

$$\leq \frac{C_\alpha}{c} \sqrt{KT}, \qquad (19)$$

where the last inequality follows from $\sqrt{x+1} - 1 \leq \sqrt{x}$ for $x > 0$. Therefore, from (16), (18), and (19), we obtain

$$\text{Reg}(T) \leq \left(\frac{8c\alpha^2(\alpha+1)e}{(\alpha-1)^2} + \frac{2\alpha^3 + (e-2)\alpha^2}{c(\alpha-1)(2\alpha-1)}\right)\sqrt{KT} + \frac{\alpha\Gamma(1-1/\alpha)}{\alpha-1} \frac{\sqrt{K}}{c},$$

which concludes the proof. $\qquad\square$

## C. Regret bound for stochastic bandits (Theorem 3.2)

To analyze the regret in the stochastic regime, we define the event

$$D_t := \left\{\sum_{i\neq i^*} \frac{1}{(2^{1/\alpha} + \eta_t \hat{\underline{L}}_{t,i})^\alpha} \leq \frac{1}{2}\right\}.$$

When $D_t$ occurs, it implies that $\underline{\hat{L}}_{t,i^*} = 0$, indicating that the optimal arm $i^*$ has been accurately identified based on the information so far. In the subsequent proof, we separately analyze the cases where $D_t$ holds and where its complement $D_t^c$ holds.

**Theorem 3.2 (restated)** *In the stochastically constrained adversarial regime with a unique best arm $i^*$, Algorithm 1 with $\alpha \in (1,3]$ and $\eta_t = cK^{\frac{1}{\alpha}-\frac{1}{2}}/\sqrt{t}$ for $c > 0$ satisfies*

$$\mathrm{Reg}(T) \leq \mathcal{O}\left(\left(\sum_{t=1}^{T}\sum_{i \neq i^*} \sqrt{K}\Delta_i^{1-\alpha}t^{-\frac{\alpha}{2}}\right) + \frac{K}{\Delta_{\min}}\right).$$

*Proof.* We bound the stability term and penalty terms by separately analyzing the contributions on the events $D_t$ and $D_t^c$.

**Stability term** For the stability term, we start from

$$\sum_{t=1}^{T}\mathbb{E}\left[\left\langle\hat{\ell}_t, \phi(\eta_t\hat{L}_t) - \phi(\eta_t\hat{L}_{t+1})\right\rangle\right] = \mathbb{E}\left[\sum_{t=1}^{T}\sum_{i\in[K]}\mathbb{E}\left[\hat{\ell}_{t,i}\left(\phi_i(\eta_t\hat{L}_t) - \phi_i(\eta_t\hat{L}_{t+1})\right)\Big|\hat{L}_t\right]\right]. \tag{20}$$

On $D_t$, we separate the contribution of the optimal arm $i^*$ from that of the suboptimal arms for a tighter analysis. For the suboptimal arms, Lemma 3.6 yields

$$\sum_{t=1}^{T}\mathbb{1}[D_t]\sum_{i\neq i^*}\mathbb{E}\left[\hat{\ell}_{t,i}\left(\phi_i(\eta_t\hat{L}_t) - \phi_i(\eta_t\hat{L}_{t+1})\right)\Big|\hat{L}_t\right] = \sum_{t=1}^{T}\mathbb{1}[D_t]\frac{cK^{\frac{1}{\alpha}-\frac{1}{2}}e(\alpha+1)}{\sqrt{t}}\sum_{j\in[K]}q_{t,j}\sum_{i\neq i^*}q_{t,i}$$

$$\leq \sum_{t=1}^{T}\mathbb{1}[D_t]\frac{cK^{\frac{1}{\alpha}-\frac{1}{2}}e(\alpha+1)}{\sqrt{t}}\sum_{j\in[K]}q_{t,j}\sum_{i\neq i^*}(2e^2w_{t,i})^{1-\frac{1}{\alpha}}.$$

Here, the last step follows from

$$q_{t,i} \leq \left(\frac{1}{1+\eta_t\underline{\hat{L}}_{t,i}}\right)^{\frac{\alpha+1}{2}} \leq (2e^2w_{t,i})^{\frac{1}{2}+\frac{1}{2\alpha}} \leq (2e^2w_{t,i})^{1-\frac{1}{\alpha}}, \tag{21}$$

where the second inequality follows from Lemma D.2, and the last one holds since $1 - \frac{1}{\alpha} \leq \frac{1}{2} + \frac{1}{2\alpha}$ for $\alpha \in (1,3]$. For the optimal arm, Lemma D.1 gives

$$\sum_{t=1}^{T}\mathbb{1}[D_t]\mathbb{E}\left[\hat{\ell}_{t,i^*}\left(\phi_{i^*}(\eta_t\hat{L}_t) - \phi_{i^*}(\eta_t\hat{L}_{t+1})\right)\Big|\hat{L}_t\right]$$

$$\leq \sum_{t=1}^{T}\mathbb{1}[D_t]\left[\sum_{j\in[K]}q_{t,j}\sum_{i\neq i^*}\frac{e(1-e^{-1})\eta_t\alpha}{(1-\zeta)^{\alpha+1}(1+\eta_t\underline{\hat{L}}_{t,i})^{\alpha+1}} + \frac{1}{1-e^{-1}}(1-e^{-1})^{\frac{\zeta}{\eta_t}}\left(\frac{\zeta}{\eta_t}+e\right)\right]$$

$$\leq \sum_{t=1}^{T}\mathbb{1}[D_t]\frac{cK^{\frac{1}{\alpha}-\frac{1}{2}}e(1-e^{-1})\alpha}{(1-\zeta)^{\alpha+1}\sqrt{t}}\sum_{j\in[K]}q_{t,j}\sum_{i\neq i^*}\left(\frac{1}{1+\eta_t\underline{\hat{L}}_{t,i}}\right)^{\frac{\alpha+1}{2}} + \mathcal{O}\left(c^2K^{\frac{2}{\alpha}-1}\right) \tag{by (37)}$$

$$\leq \sum_{t=1}^{T}\mathbb{1}[D_t]\frac{cK^{\frac{1}{\alpha}-\frac{1}{2}}e(1-e^{-1})\alpha}{(1-\zeta)^{\alpha+1}\sqrt{t}}\sum_{j\in[K]}q_{t,j}\sum_{i\neq i^*}(2e^2w_{t,i})^{1-\frac{1}{\alpha}} + \mathcal{O}\left(c^2K^{\frac{2}{\alpha}-1}\right). \tag{by (21)}$$

Combining the contributions from both the optimal and suboptimal arms, we obtain

$$\sum_{t=1}^{T} \mathbb{1}[D_t] \sum_{i \in [K]} \mathbb{E}\left[\hat{\ell}_{t,i}\left(\phi_i(\eta_t \hat{L}_t) - \phi_i(\eta_t \hat{L}_{t+1})\right)\Big|\hat{L}_t\right]$$

$$= \sum_{t=1}^{T} \mathbb{1}[D_t]\left(\alpha + 1 + \frac{\alpha(1 - e^{-1})}{(1-\zeta)^{\alpha+1}}\right)\frac{cK^{\frac{1}{\alpha}-\frac{1}{2}}e}{\sqrt{t}} \sum_{j \in [K]} q_{t,j} \sum_{i \neq i^*}(2e^2 w_{t,i})^{1-\frac{1}{\alpha}} + \mathcal{O}\left(c^2 K^{\frac{2}{\alpha}-1}\right)$$

$$\leq \sum_{t=1}^{T} \mathbb{1}[D_t]\left(\frac{2c\alpha(\alpha+1)e}{\alpha-1} + \frac{2c\alpha^2(1-e^{-1})e}{(\alpha-1)(1-\zeta)^{\alpha+1}}\right)\frac{K^{\frac{1}{2\alpha}}}{\sqrt{t}} \sum_{i \neq i^*}(2e^2 w_{t,i})^{1-\frac{1}{\alpha}} + \mathcal{O}\left(c^2 K^{\frac{2}{\alpha}-1}\right),$$

where the last step follows from (17). On $D_t^c$, we have

$$\sum_{t=1}^{T} \mathbb{1}[D_t^c]\mathbb{E}\left[\left\langle \hat{\ell}_t, \phi(\eta_t \hat{L}_t) - \phi(\eta_t \hat{L}_{t+1})\right\rangle\Big|\hat{L}_t\right] = \sum_{t=1}^{T} \mathbb{1}[D_t^c]\frac{cK^{\frac{1}{\alpha}-\frac{1}{2}}e(\alpha+1)}{\sqrt{t}} \sum_{j \in [K]} q_{t,j} \sum_{i \in [K]} q_{t,i}$$

$$\leq \sum_{t=1}^{T} \mathbb{1}[D_t^c]\frac{4\alpha^2(\alpha+1)e}{(\alpha-1)^2}\frac{cK^{\frac{1}{\alpha}-\frac{1}{2}}}{\sqrt{t}}K^{1-\frac{1}{\alpha}} \qquad \text{(by (17))}$$

$$= \sum_{t=1}^{T} \mathbb{1}[D_t^c]\frac{4c\alpha^2(\alpha+1)e}{(\alpha-1)^2}\sqrt{\frac{K}{t}}.$$

Combining the bounds for both $D_t$ and $D_t^c$, the stability term can be bounded as

$$\sum_{t=1}^{T} \mathbb{E}\left[\left\langle \hat{\ell}_t, \phi(\eta_t \hat{L}_t) - \phi(\eta_t \hat{L}_{t+1})\right\rangle\right]$$

$$\leq \mathbb{E}\left[\sum_{t=1}^{T} \mathbb{1}[D_t]\left(\frac{2c\alpha(\alpha+1)e}{\alpha-1} + \frac{2c\alpha^2(1-e^{-1})e}{(\alpha-1)(1-\zeta)^{\alpha+1}}\right)\frac{K^{\frac{1}{2\alpha}}}{\sqrt{t}} \sum_{i \neq i^*}(2e^2 w_{t,i})^{1-\frac{1}{\alpha}}\right]$$

$$+ \mathbb{E}\left[\sum_{t=1}^{T} \mathbb{1}[D_t^c]\frac{4c\alpha^2(\alpha+1)e}{(\alpha-1)^2}\sqrt{\frac{K}{t}}\right] + \mathcal{O}\left(c^2 K^{\frac{2}{\alpha}-1}\right). \tag{22}$$

**Penalty term** For the penalty term, we can start from (18):

$$\sum_{t=1}^{T+1}\left(\frac{1}{\eta_t} - \frac{1}{\eta_{t-1}}\right)\mathbb{E}_{r_t \sim \mathcal{P}_\alpha^K}\left[r_{t,i_t} - r_{t,i^*}\right]$$

$$\leq \sum_{t=2}^{T+1}\left(\frac{1}{\eta_t} - \frac{1}{\eta_{t-1}}\right)\mathbb{E}_{r_t \sim \mathcal{P}_\alpha^K}\left[r_{t,i_t} - r_{t,i^*}\right] + \frac{\alpha\Gamma(1-1/\alpha)}{\alpha-1}\frac{\sqrt{K}}{c}. \tag{23}$$

On $D_t$, we obtain

$$\sum_{t=2}^{T+1} \mathbb{1}[D_t]\left(\frac{1}{\eta_t} - \frac{1}{\eta_{t-1}}\right)\mathbb{E}_{r_t \sim \mathcal{P}_\alpha^K}\left[r_{t,i_t} - r_{t,i^*}\Big|\hat{L}_t\right]$$

$$\leq \sum_{t=2}^{T+1} \mathbb{1}[D_t]\frac{\alpha}{\alpha-1}\left(\frac{1}{\eta_t} - \frac{1}{\eta_{t-1}}\right)\sum_{i \neq i^*}\frac{1}{(1+\eta_t \hat{\underline{L}}_{t,i})^{\alpha-1}}. \qquad \text{(by Lemma 3.5)}$$

For $t \geq 2$, Lemma D.2 implies

$$\left(\frac{1}{\eta_t} - \frac{1}{\eta_{t-1}}\right)\sum_{i \neq i^*}\frac{1}{(1+\eta_t \hat{\underline{L}}_{t,i})^{\alpha-1}} \leq \frac{K^{\frac{1}{2}-\frac{1}{\alpha}}}{2c\sqrt{t-1}} \sum_{i \neq i^*}(2e^2 w_{t,i})^{1-\frac{1}{\alpha}},$$

which yields

$$\sum_{t=2}^{T+1} \mathbb{1}[D_t]\left(\frac{1}{\eta_t} - \frac{1}{\eta_{t-1}}\right)\mathbb{E}_{r_t \sim \mathcal{P}_\alpha^K}\left[r_{t,i_t} - r_{t,i^*}\Big|\hat{L}_t\right] \le \sum_{t=1}^{T} \mathbb{1}[D_t]\frac{\alpha}{2c(\alpha-1)}\frac{K^{\frac{1}{2\alpha}}}{\sqrt{t}}\sum_{i \ne i^*}(2e^2 w_{t,i})^{1-\frac{1}{\alpha}},$$

where we used the fact that $\frac{1}{2} - \frac{1}{\alpha} \le \frac{1}{2\alpha}$ for $\alpha \in (1,3]$.

On $D_t^c$, we obtain

$$\begin{aligned}
\sum_{t=2}^{T+1} \mathbb{1}[D_t^c]\left(\frac{1}{\eta_t} - \frac{1}{\eta_{t-1}}\right)\mathbb{E}_{r_t \sim \mathcal{P}_\alpha^K}\left[r_{t,i_t} - r_{t,i^*}\Big|\hat{L}_t\right] &\le \sum_{t=2}^{T} \mathbb{1}[D_t^c]\left(\frac{1}{\eta_t} - \frac{1}{\eta_{t-1}}\right)C_\alpha K^{\frac{1}{\alpha}} && \text{(by Lemma 3.5)}\\
&\le \sum_{t=1}^{T} \mathbb{1}[D_t^c]\frac{K^{\frac{1}{2}-\frac{1}{\alpha}}}{c\sqrt{2}\sqrt{t}}C_\alpha K^{\frac{1}{\alpha}}\\
&= \sum_{t=1}^{T} \mathbb{1}[D_t^c]\frac{C_\alpha}{c\sqrt{2}}\sqrt{\frac{K}{t}},
\end{aligned}$$

where the first step assumes $\eta_T = \eta_{T+1}$ for simplicity and the second step is due to the fact that $\sqrt{t/(t-1)} \le \sqrt{2}$ for $t \ge 2$. Here, the assumption $\eta_T = \eta_{T+1}$ does not affect the behavior of the algorithm, as the procedure terminates at round $T$. Although this assumes knowledge of $T$, even without it one can just introduce an additional $\mathcal{O}(1/\sqrt{T+1})$ term, which does not affect the overall regret for sufficiently large $T$. Combining the bounds for both $D_t$ and $D_t^c$, the penalty term can be bounded as

$$\begin{aligned}
&\sum_{t=1}^{T+1}\left(\frac{1}{\eta_t} - \frac{1}{\eta_{t-1}}\right)\mathbb{E}_{r_t \sim \mathcal{P}_\alpha^K}[r_{t,i_t} - r_{t,i^*}]\\
&\le \mathbb{E}\left[\sum_{t=1}^{T} \mathbb{1}[D_t]\frac{\alpha}{2c(\alpha-1)}\frac{K^{\frac{1}{2\alpha}}}{\sqrt{t}}\sum_{i \ne i^*}(2e^2 w_{t,i})^{1-\frac{1}{\alpha}} + \sum_{t=1}^{T} \mathbb{1}[D_t^c]\frac{C_\alpha}{c\sqrt{2}}\sqrt{\frac{K}{t}}\right] + \frac{\alpha\Gamma(1-1/\alpha)}{\alpha-1}\frac{\sqrt{K}}{c}. \quad (24)
\end{aligned}$$

Finally, combining (22) with (24), the regret can be upper bounded as

$$\begin{aligned}
\text{Reg}(T) &\le \mathbb{E}\left[\sum_{t=1}^{T} \mathbb{1}[D_t]\left(\frac{2c\alpha(\alpha+1)e}{\alpha-1} + \frac{2c\alpha^2(1-e^{-1})e}{(\alpha-1)(1-\zeta)^{\alpha+1}} + \frac{\alpha}{2c(\alpha-1)}\right)\frac{K^{\frac{1}{2\alpha}}}{\sqrt{t}}\sum_{i \ne i^*}(2e^2 w_{t,i})^{1-\frac{1}{\alpha}}\right]\\
&\quad + \mathbb{E}\left[\sum_{t=1}^{T} \mathbb{1}[D_t^c]\left(\frac{4c\alpha^2(\alpha+1)e}{(\alpha-1)^2} + \frac{C_\alpha}{c\sqrt{2}}\right)\sqrt{\frac{K}{t}}\right] + \mathcal{O}\left(c^2 K^{\frac{2}{\alpha}-1}\right) + \frac{\alpha\Gamma(1-1/\alpha)}{\alpha-1}\frac{\sqrt{K}}{c}. \quad (25)
\end{aligned}$$

**Self-bounding technique**  We employ the self-bounding technique of Zimmert & Seldin (2021) in the stochastically constrained adversarial regime to demonstrate that our policy adapts to broader settings beyond the purely stochastic regime.

Specifically,

$$
\begin{aligned}
\mathrm{Reg}(T) &= 2 \cdot \mathrm{Reg}(T) - \mathbb{E}\left[\sum_{t=1}^{T} \sum_{i \neq i^*} \Delta_i w_{t,i}\right] && \text{(by (2))} \\
&= 2 \cdot \mathrm{Reg}(T) - \mathbb{E}\left[\sum_{t=1}^{T} \left(\mathbb{I}[D_t] \sum_{i \neq i^*} \Delta_i w_{t,i} + \mathbb{I}[D_t^c] \sum_{i \neq i^*} \Delta_i w_{t,i}\right)\right] \\
&\leq \mathcal{O}\left(c^2 K^{\frac{2}{\alpha}-1}\right) + \frac{2\alpha\Gamma(1-1/\alpha)}{\alpha-1}\frac{\sqrt{K}}{c} + \mathbb{E}\left[\sum_{t=1}^{T} \mathbb{I}[D_t] \sum_{i \neq i^*} \left(\frac{Z_1(\alpha)w_{t,i}^{1-\frac{1}{\alpha}}}{\sqrt{t}} - \Delta_i w_{t,i}\right)\right] \\
&\quad + \mathbb{E}\left[\sum_{t=1}^{T} \mathbb{I}[D_t^c]\left(\left(\frac{8c\alpha^2(\alpha+1)e}{(\alpha-1)^2} + \frac{C_\alpha\sqrt{2}}{c}\right)\sqrt{\frac{K}{t}} - \frac{1-e^{-1/2}}{2}\Delta_{\min}\right)\right] && \text{(by (D.3))} \\
&\leq \mathcal{O}\left(c^2 K^{\frac{2}{\alpha}-1}\right) + \frac{2\alpha\Gamma(1-1/\alpha)}{\alpha-1}\frac{\sqrt{K}}{c} + \mathbb{E}\left[\sum_{t=1}^{T} \mathbb{1}[D_t] \sum_{i \neq i^*} Z_2(\alpha)\Delta_i^{1-\alpha}t^{-\frac{\alpha}{2}}\right] \\
&\quad + \mathbb{E}\left[\sum_{t=1}^{T} \mathbb{1}[D_t^c]\max\left(\left(\frac{8c\alpha^2(\alpha+1)e}{(\alpha-1)^2} + \frac{C_\alpha\sqrt{2}}{c}\right)\sqrt{\frac{K}{t}} - \frac{1-e^{-1/2}}{2}\Delta_{\min}, 0\right)\right], && \text{(26)}
\end{aligned}
$$

where the last step follows from

$$
\sum_{i \neq i^*} \left(\frac{Z_1(\alpha)w_{t,i}^{1-\frac{1}{\alpha}}}{\sqrt{t}} - \Delta_i w_{t,i}\right) \leq \sum_{i \neq i^*} \max_{w \in [0,1]} \left(\frac{Z_1(\alpha)w^{1-\frac{1}{\alpha}}}{\sqrt{t}} - \Delta_i w\right) \leq \sum_{i \neq i^*} Z_2(\alpha)\Delta_i^{1-\alpha}t^{-\frac{\alpha}{2}},
$$

which is a direct application of Lemma 8 in Rouyer & Seldin (2020). Here, the constants $Z_1(\alpha;\zeta)$ and $Z_2(\alpha;\zeta)$ are

$$
Z_1(\alpha;\zeta) = Z_1(\alpha) = (2e^2)^{1-\frac{1}{\alpha}}\left(\frac{4c\alpha(\alpha+1)e}{\alpha-1} + \frac{4c\alpha^2(1-e^{-1})e}{(\alpha-1)(1-\zeta)^{\alpha+1}} + \frac{\alpha}{c(\alpha-1)}\right)K^{\frac{1}{2\alpha}} \tag{27}
$$

and

$$
Z_2(\alpha;\zeta) = Z_2(\alpha) = Z_1(\alpha)^\alpha\left(\left(\frac{\alpha-1}{\alpha}\right)^{\alpha-1} - \left(\frac{\alpha-1}{\alpha}\right)^\alpha\right). \tag{28}
$$

Next, we define the time step $T_{\mathrm{cut}}$ such that for $t > \lfloor T_{\mathrm{cut}}\rfloor$, the last term in (26) evaluates to zero. Hence, we can bound the sum as

$$
\sum_{t=1}^{T} \max\left(A\sqrt{\frac{K}{t}} - B\cdot\Delta_{\min}, 0\right) \leq \sum_{t=1}^{\lfloor T_{\mathrm{cut}}\rfloor}\left(A\sqrt{\frac{K}{t}} - B\cdot\Delta_{\min}\right) \leq \frac{2A^2}{B}\frac{K}{\Delta_{\min}}, \tag{29}
$$

where $T_{\mathrm{cut}} := \frac{A^2 K}{B^2 \Delta_{\min}^2}$. Using this, we can upper bound (26) as

$$
\begin{aligned}
&\mathcal{O}\left(c^2 K^{\frac{2}{\alpha}-1}\right) + \frac{2\alpha\Gamma(1-1/\alpha)}{\alpha-1}\frac{\sqrt{K}}{c} + \mathbb{E}\left[\sum_{t=1}^{T} \mathbb{1}[D_t] \sum_{i \neq i^*} Z_2(\alpha)\Delta_i^{1-\alpha}t^{-\frac{\alpha}{2}}\right] \\
&\quad + \mathbb{E}\left[\sum_{t=1}^{T} \mathbb{1}[D_t^c]\max\left(\left(\frac{8c\alpha^2(\alpha+1)e}{(\alpha-1)^2} + \frac{C_\alpha\sqrt{2}}{c}\right)\sqrt{\frac{K}{t}} - \frac{1-e^{-1/2}}{2}\Delta_{\min}, 0\right)\right] \\
&\leq \mathcal{O}\left(c^2 K^{\frac{2}{\alpha}-1}\right) + \frac{2\alpha\Gamma(1-1/\alpha)}{\alpha-1}\frac{\sqrt{K}}{c} + \sum_{t=1}^{T}\sum_{i \neq i^*} Z_2(\alpha)\Delta_i^{1-\alpha}t^{-\frac{\alpha}{2}} \\
&\quad + \frac{4}{1-e^{-1/2}}\left(\frac{8c\alpha^2(\alpha+1)e}{(\alpha-1)^2} + \frac{C_\alpha\sqrt{2}}{c}\right)^2 \frac{K}{\Delta_{\min}}, && \text{(by (29))}
\end{aligned}
$$

which implies

$$\text{Reg}(T) \leq \mathcal{O}\left(\left(\sum_{t=1}^{T}\sum_{i\neq i^*}\sqrt{K}\Delta_i^{1-\alpha}t^{-\frac{\alpha}{2}}\right) + \frac{K}{\Delta_{\min}}\right),$$

and thus concludes the proof. □

**Corollary 3.3 (restated)** *In the stochastically constrained adversarial regime with a unique best arm $i^*$, Algorithm 1 with $\alpha = 3$ and $\eta_t = cK^{\frac{1}{\alpha}-\frac{1}{2}}/\sqrt{t}$ for $c > 0$ satisfies*

$$\text{Reg}(T) \leq \mathcal{O}\left(\sqrt{\frac{K}{\Delta_{\min}}\sum_{i\neq i^*}\frac{1}{\Delta_i}} + \frac{K}{\Delta_{\min}}\right).$$

*Proof.* By Theorem 3.2, for $\alpha \in (1, 3]$, the regret of our policy satisfies

$$\text{Reg}(T) \leq \mathcal{O}\left(\left(\sum_{t=1}^{T}\sum_{i\neq i^*}\sqrt{K}\Delta_i^{1-\alpha}t^{-\frac{\alpha}{2}}\right) + \frac{K}{\Delta_{\min}}\right),$$

which depends on the time horizon $T$. Inspired by Theorem 4 of Rouyer & Seldin (2020), we next derive a $T$-independent bound for $\alpha \in (2, 3]$ and identify the value of $\alpha$ that minimizes this bound.

Let $T_0 := D\left(\frac{x}{\Delta_{\min}}\right)^2$ for $D \geq 1$ and $x \in \mathbb{R}_{>0}$. For $t \leq T_0$, the proof follows identically to the adversarial regime, yielding a contribution of order $\mathcal{O}(\sqrt{KT_0})$. For the remaining rounds $t > T_0$, we apply the same argument as in the stochastic case, i.e., (25), which gives

$$\begin{aligned}
\text{Reg}(T) \leq{}& \mathbb{E}\left[\sum_{t=T_0+1}^{T}\mathbb{1}[D_t]\left(\frac{2c\alpha(\alpha+1)e}{\alpha-1} + \frac{2c\alpha^2(1-e^{-1})e}{(\alpha-1)(1-\zeta)^{\alpha+1}} + \frac{\alpha}{2c(\alpha-1)}\right)\frac{K^{\frac{1}{2\alpha}}}{\sqrt{t}}\sum_{i\neq i^*}(2e^2 w_{t,i})^{1-\frac{1}{\alpha}}\right] \\
&+ \mathbb{E}\left[\sum_{t=T_0+1}^{T}\mathbb{1}[D_t^c]\left(\frac{4c\alpha^2(\alpha+1)e}{(\alpha-1)^2} + \frac{C_\alpha}{c\sqrt{2}}\right)\sqrt{\frac{K}{t}}\right] + \left(\frac{8c\alpha^2(\alpha+1)e}{(\alpha-1)^2} + \frac{C_\alpha}{c}\right)\sqrt{KT_0} \\
&+ \mathcal{O}\left(c^2 K^{\frac{2}{\alpha}-1}\right) + \frac{\alpha\Gamma(1-1/\alpha)}{\alpha-1}\frac{\sqrt{K}}{c}, \qquad\qquad (30)
\end{aligned}$$

for any $\zeta \in (0, 1)$ and $\alpha \in (1, 3]$. Here, $C_\alpha = \frac{2\alpha^3+(e-2)\alpha^2}{(\alpha-1)(2\alpha-1)}$ denotes the constant defined in Lemma 3.5. Following the

similar steps as Theorem 3.2, we obtain

$$
\text{Reg}(T) = 2 \cdot \text{Reg}(T) - \mathbb{E}\left[\sum_{t=1}^{T}\sum_{i \neq i^*} \Delta_i w_{t,i}\right] \tag{31}
$$

$$
\leq 2 \cdot \text{Reg}(T) - \mathbb{E}\left[\sum_{t=T_0+1}^{T}\sum_{i \neq i^*} \Delta_i w_{t,i}\right]
$$

$$
\leq \mathbb{E}\left[\sum_{t=T_0+1}^{T} \mathbb{1}[D_t]\sum_{i \neq i^*}\left(\frac{Z_1(\alpha)w_{t,i}^{1-\frac{1}{\alpha}}}{\sqrt{t}} - \Delta_i w_{t,i}\right)\right]
$$

$$
+ \mathbb{E}\left[\sum_{t=T_0+1}^{T} \mathbb{1}[D_t^c]\left(\left(\frac{8c\alpha^2(\alpha+1)e}{(\alpha-1)^2} + \frac{C_\alpha\sqrt{2}}{c}\right)\sqrt{\frac{K}{t}} - \frac{1-e^{-1/2}}{2}\Delta_{\min}\right)\right]
$$

$$
+ \left(\frac{16c\alpha^2(\alpha+1)e}{(\alpha-1)^2} + \frac{2C_\alpha}{c}\right)\sqrt{KT_0} + \mathcal{O}\left(c^2 K^{\frac{2}{\alpha}-1}\right) + \frac{2\alpha\Gamma(1-1/\alpha)}{\alpha-1}\frac{\sqrt{K}}{c}
$$

$$
\leq \sum_{t=T_0+1}^{T}\sum_{i \neq i^*} Z_2(\alpha)\Delta_i^{1-\alpha}t^{-\frac{\alpha}{2}} + \frac{4}{1-e^{-1/2}}\left(\frac{8c\alpha^2(\alpha+1)e}{(\alpha-1)^2} + \frac{C_\alpha\sqrt{2}}{c}\right)^2\frac{K}{\Delta_{\min}}
$$

$$
+ \left(\frac{16c\alpha^2(\alpha+1)e}{(\alpha-1)^2} + \frac{2C_\alpha}{c}\right)\sqrt{KT_0} + \mathcal{O}\left(c^2 K^{\frac{2}{\alpha}-1}\right) + \frac{2\alpha\Gamma(1-1/\alpha)}{\alpha-1}\frac{\sqrt{K}}{c}. \tag{32}
$$

Here, $Z_1(\alpha)$ and $Z_2(\alpha)$ are the constants defined in (27) and (28), respectively. By Lemma D.4, for $\alpha \in (2,3]$, the first term in (32) can be bounded as

$$
\sum_{t=T_0+1}^{T}\sum_{i \neq i^*} Z_2(\alpha)\Delta_i^{1-\alpha}t^{-\frac{\alpha}{2}} \leq \sum_{i \neq i^*} Z_3(\alpha)\frac{\sqrt{K}D^{1-\frac{\alpha}{2}}}{\Delta_i},
$$

which gives

$$
\text{Reg}(T) \leq \sum_{i \neq i^*} Z_3(\alpha)\frac{\sqrt{K}D^{1-\frac{\alpha}{2}}}{\Delta_i} + \frac{4}{1-e^{-1/2}}\left(\frac{8c\alpha^2(\alpha+1)e}{(\alpha-1)^2} + \frac{C_\alpha\sqrt{2}}{c}\right)^2\frac{K}{\Delta_{\min}}
$$

$$
+ \left(\frac{16c\alpha^2(\alpha+1)e}{(\alpha-1)^2} + \frac{2C_\alpha}{c}\right)\sqrt{KT_0} + \mathcal{O}\left(c^2 K^{\frac{2}{\alpha}-1}\right) + \frac{2\alpha\Gamma(1-1/\alpha)}{\alpha-1}\frac{\sqrt{K}}{c}
$$

$$
= \sum_{i \neq i^*} Z_3(\alpha)\frac{\sqrt{K}D^{1-\frac{\alpha}{2}}}{\Delta_i} + \frac{4}{1-e^{-1/2}}\left(\frac{8c\alpha^2(\alpha+1)e}{(\alpha-1)^2} + \frac{C_\alpha\sqrt{2}}{c}\right)^2\frac{K}{\Delta_{\min}}
$$

$$
+ \left(\frac{16c\alpha^2(\alpha+1)e}{(\alpha-1)^2} + \frac{2C_\alpha}{c}\right)\frac{x\sqrt{KD}}{\Delta_{\min}} + \mathcal{O}\left(c^2 K^{\frac{2}{\alpha}-1}\right) + \frac{2\alpha\Gamma(1-1/\alpha)}{\alpha-1}\frac{\sqrt{K}}{c}, \tag{33}
$$

where

$$
Z_3(\alpha) = \frac{2x^{2-\alpha}Z_2(\alpha)}{(\alpha-2)\sqrt{K}}.
$$

The first and third terms in (33) are minimized when $\alpha = 3$ and

$$
D = \frac{Z_3(3)}{x(144ce + 2C_3/c)}\sum_{i \neq i^*}\frac{\Delta_{\min}}{\Delta_i},
$$

which, by the AM-GM inequality, leads to the bound

$$\sum_{i\neq i^*} \frac{Z_3(3)}{\Delta_i} \frac{\sqrt{K}}{\sqrt{D}} + \left(144ce + \frac{2C_3}{c}\right)\frac{x\sqrt{KD}}{\Delta_{\min}} \leq 2\sqrt{\left(144ce + \frac{2C_3}{c}\right)\frac{xZ_3(3)K}{\Delta_{\min}}\sum_{i\neq i^*}\frac{1}{\Delta_i}} \tag{34}$$

$$\leq \mathcal{O}\left(\sqrt{\frac{K}{\Delta_{\min}}\sum_{i\neq i^*}\frac{1}{\Delta_i}}\right). \tag{35}$$

Note that the feasible range of $x \in \mathbb{R}_{>0}$ is determined by the two constraints $T_0 \leq T_{\text{cut}}$ and $D \geq 1$:

$$x \leq \min\left\{\frac{4(144ce + 2C_3/c)(72ce + C_3\sqrt{2}/c)^2}{Z_3(3)(1 - e^{-1/2})^2}\frac{K}{\Delta_{\min}}\left(\sum_{i\neq i^*}\frac{1}{\Delta_i}\right)^{-1}, \frac{Z_3(3)}{144ce + 2C_3/c}\sum_{i\neq i^*}\frac{\Delta_{\min}}{\Delta_i}\right\}.$$

Any choice of $x$ within this range is valid, since $x$ cancels out in (34) through the term $xZ_3(3)$, as $Z_3(3)$ contains $x^{-1}$, and thus does not affect the final bound. Finally, from (33) and (35), we obtain

$$\text{Reg}(T) \leq \mathcal{O}\left(\sqrt{\frac{K}{\Delta_{\min}}\sum_{i\neq i^*}\frac{1}{\Delta_i}} + \frac{K}{\Delta_{\min}}\right),$$

which concludes the proof.

**The choice of $c$** We determine the choice of $c$ in (34). Specifically,

$$\left(144ce + \frac{2C_3}{c}\right)xZ_3(3) = \frac{32e^4}{27}\left(144ce + \frac{36 + 9e}{5c}\right)\left(24ce + \frac{18ce(1 - e^{-1})}{(1 - \zeta)^4} + \frac{3}{2c}\right)^3. \tag{36}$$

For notational simplicity, we express the RHS in the form $f(c) = (h_1c + h_2/c)(h_3c + h_4/c)^3$, where $h_1, h_2, h_3, h_4$ are constants determined by the coefficients above. To minimize $f(c)$, we substitute $x = c^2$ and define $g(x) = (h_1x+h_2)(h_3x+h_4)^3/x^2$. Differentiating $g(x)$ with respect to $x$ gives

$$g'(x) = \frac{(h_3x + h_4)^2(2h_1h_3x^2 + (h_2h_3 - h_1h_4)x - 2h_2h_4)}{x^3}.$$

Thus, the stationary points of $g(x)$, where $g'(x) = 0$, are obtained by solving the quadratic equation $2h_1h_3x^2 + (h_2h_3 - h_1h_4)x - 2h_2h_4 = 0$. This yields

$$x^* = \frac{-h_2h_3 + h_1h_4 + \sqrt{(h_2h_3 - h_1h_4)^2 + 16h_1h_2h_3h_4}}{4h_1h_3}.$$

Recalling that $x = c^2$, the optimal choice of $c$ is given by

$$c^* = \sqrt{x^*}.$$

When $\zeta = 10^{-1}$, the above computation gives

$$c^* \simeq 0.128, \quad f(c) \simeq 25799.360.$$

As $\zeta$ decreases, the resulting constant decreases.

*Remark* C.1. There are some possible ways to further reduce the leading multiplicative constant of (36) in the regret bound. Firstly, in (31), one may introduce a parameter $\kappa \in (0, 1)$ and consider $(30) - \kappa \times (2)$. In our analysis, we use $\kappa = 1/2$ for simplicity, but optimizing over $\kappa$ can reduce the constant. Secondly, in (10), one may define

$$D_t = \left\{\sum_{i\neq i^*}\frac{1}{(y^{1/\alpha} + \eta_t\hat{\underline{L}}_{t,i})^\alpha} \leq \frac{1}{y}\right\},$$

where we set $y = 2$ for simplicity in our analysis. By optimizing $y$, the constant in the regret bound can be reduced. For instance, the term $2e^2$ in (21) contributes to the constant, and under the above definition of $D_t$, it becomes $ye^{\frac{2}{y-1}}$, which can be decreased by choosing an appropriate $y$. Finally, in the process of choosing $c$, we set $\zeta = 10^{-1}$ for simplicity. Together with the optimizations discussed above, tuning $\zeta$ could further reduce the constant in the regret bound.

$\square$

## D. Auxiliary lemmas

Here, we include several lemmas, along with their proofs if necessary, that are used in the appendix.

**Lemma D.1** (modified Lemma 25 of Lee et al. (2024)). *On $D_t$, for any $\zeta \in (0, 1)$, FTPL with Pareto perturbations of shape $\alpha > 1$ satisfies*

$$
\mathbb{E}\left[\hat{\ell}_{t,i^*}\left(\phi_{i^*}(\eta_t \hat{L}_t) - \phi_{i^*}(\eta_t \hat{L}_{t+1})\right)\Big|\hat{L}_t\right]
$$
$$
\leq \sum_{j\in[K]} q_{t,j} \cdot \sum_{i\neq i^*} \frac{e(1-e^{-1})\eta_t\alpha}{(1-\zeta)^{\alpha+1}(1+\eta_t\underline{\hat{L}}_{t,i})^{\alpha+1}} + \frac{1}{1-e^{-1}}(1-e^{-1})^{\frac{\zeta}{\eta_t}}\left(\frac{\zeta}{\eta_t}+e\right),
$$

*and when $\eta_t = cK^{\frac{1}{\alpha}-\frac{1}{2}}/\sqrt{t}$,*

$$
\sum_{t=1}^{\infty} \frac{1}{1-e^{-1}}(1-e^{-1})^{\frac{\zeta}{\eta_t}}\left(\frac{\zeta}{\eta_t}+e\right) \leq \mathcal{O}\left(c^2 K^{\frac{2}{\alpha}-1}\right). \tag{37}
$$

*Proof.* Note that our formulation differs slightly from that in Lee et al. (2024), as we require a tighter result for the later use of this lemma. Nevertheless, the overall proof remains almost the same and thus we only provide details for the parts that differ.

As the proof in Honda et al. (2023) and Lee et al. (2024), we consider two cases (i) $p_{t,i^*}^{-1} \leq \zeta/\eta_t$ and (ii) $p_{t,i^*}^{-1} > \zeta/\eta_t$ separately. Notice that the case (ii) can be directly obtained by Lemma 11 in Honda et al. (2023) (or Lemma 23 in Lee et al. (2024)), which shows that

$$
\mathbb{E}\left[\mathbb{1}\left[\hat{\ell}_{t,i^*} > \frac{\zeta}{\eta_t}\right]\hat{\ell}_{t,i^*}\Big|\hat{L}_t\right] \leq \frac{1}{1-e^{-1}}(1-e^{-1})^{\frac{\zeta}{\eta_t}}\left(\frac{\zeta}{\eta_t}+e\right).
$$

When $p_{t,i}^{-1} \leq \zeta/\eta_t$, on $D_t$ (where $\hat{L}_{t,i^*} = 0$), we have

$$
\phi_{i^*}(\eta_t(\hat{L}_t + xe_{i^*})) = \int_1^{\infty} f(z) \prod_{j\neq i^*} F\left(z + \eta_t(\underline{\hat{L}}_{t,j} - x)\right)\mathrm{d}z.
$$

This implies that for $x \leq \frac{\zeta}{\eta_t}$

$$-\frac{\mathrm{d}}{\mathrm{d}x}\phi_{i^*}(\eta_t(\hat{L}_t + xe_{i^*}))$$

$$= \int_1^\infty f(z) \sum_{i \neq i^*} \left( \eta_t f\left(z + \eta_t(\hat{\underline{L}}_{t,i} - x)\right) \prod_{j \neq i, i^*} \left(1 - F\left(z + \eta_t(\hat{\underline{L}}_{t,j} - x)\right)\right) \right) \mathrm{d}z$$

$$\leq \int_1^\infty f(z) \sum_{i \neq i^*} \left( \eta_t f\left(z + \eta_t(\hat{\underline{L}}_{t,i} - x)\right) \exp\left(-\sum_{j \neq i, i^*}\left(1 - F\left(z + \eta_t(\hat{\underline{L}}_{t,j} - x)\right)\right)\right) \right) \mathrm{d}z$$

$$\leq e \int_1^\infty f(z) \sum_{i \neq i^*} \left( \eta_t f\left(z + \eta_t(\hat{\underline{L}}_{t,i} - x)\right) \exp\left(-\sum_{j \neq i, i^*}\left(1 - F\left(z + \eta_t(\hat{\underline{L}}_{t,j} - x)\right)\right) - (1 - F(z))\right) \right) \mathrm{d}z$$

$$\leq e \int_1^\infty f(z) \sum_{i \neq i^*} \eta_t f\left(z + \eta_t(\hat{\underline{L}}_{t,i} - x)\right) \exp(-(1 - F(z)))\mathrm{d}z$$

$$= e \int_1^\infty f(z) \sum_{i \neq i^*} \eta_t \frac{\alpha}{(z + \eta_t(\hat{\underline{L}}_{t,i} - x))^{\alpha+1}} \exp(-(1 - F(z)))\mathrm{d}z$$

$$\leq \sum_{i \neq i^*} \frac{e\eta_t\alpha}{(1 - \zeta)^{\alpha+1}(1 + \eta_t\hat{\underline{L}}_{t,i})^{\alpha+1}} \int_1^\infty f(z)\exp(-(1 - F(z)))\mathrm{d}z \tag{38}$$

$$= \sum_{i \neq i^*} \frac{e(1 - e^{-1})\eta_t\alpha}{(1 - \zeta)^{\alpha+1}(1 + \eta_t\hat{\underline{L}}_{t,i})^{\alpha+1}}, \tag{39}$$

where (38) follows from $x \leq \zeta/\eta_t$ and

$$\frac{1}{(z + a - b)} \leq \frac{1}{(1 + a)(1 - b)}, \quad \forall z \geq 1, b < 1, a \geq 0.$$

Therefore, we have

$$\mathbb{E}\left[\mathbb{1}[\hat{\ell}_{t,i^*} \leq \zeta/\eta_t]\hat{\ell}_{t,i^*}\left(\phi_{i^*}(\eta_t\hat{L}_t) - \phi_{i^*}(\eta_t\hat{L}_{t+1})\right)\Big|\hat{L}_t\right]$$

$$\leq \mathbb{E}\left[\mathbb{1}[\hat{\ell}_{t,i} \leq \zeta/\eta_t]\hat{\ell}_{t,i^*}^2 \sum_{i \neq i^*} \frac{e(1 - e^{-1})\eta_t\alpha}{(1 - \zeta)^{\alpha+1}(1 + \eta_t\hat{\underline{L}}_{t,i})^{\alpha+1}}\Big|\hat{L}_t\right] \quad \text{(by (39))}$$

$$\leq \mathbb{E}\left[\frac{\ell_{t,i^*}^2}{p_{t,i^*}} \sum_{i \neq i^*} \frac{e(1 - e^{-1})\eta_t\alpha}{(1 - \zeta)^{\alpha+1}(1 + \eta_t\hat{\underline{L}}_{t,i})^{\alpha+1}}\Big|\hat{L}_t\right]$$

$$\leq \sum_{j \in [K]} q_{t,j} \sum_{i \neq i^*} \frac{e(1 - e^{-1})\eta_t\alpha}{(1 - \zeta)^{\alpha+1}(1 + \eta_t\hat{\underline{L}}_{t,i})^{\alpha+1}},$$

where the last inequality follows from $\ell_t \in [0, 1]^K$ and $p_{t,i^*} = \frac{q_{t,i^*}}{\sum_{j \in [K]} q_{t,j}}$ with $q_{t,i^*} = 1$ on $D_t$. $\qquad\square$

**Lemma D.2.** *For FTPL with Pareto perturbations with shape $\alpha > 1$, it holds that*

$$w_{t,i} \leq \frac{1}{(1 + \eta_t\hat{\underline{L}}_{t,i})^\alpha}, \forall t \in [T], i \in [K] \quad \text{and} \quad w_{t,i} \geq \frac{1}{2e^2(1 + \eta_t\hat{\underline{L}}_{t,i})^\alpha} \text{ on } D_t, \forall i \neq i^*.$$

*In addition, the optimal arm satisfies $w_{t,i^*} \geq \frac{1}{2e}$ on $D_t$.*

*Proof.* By definition of $w_{t,i}$, for any $i \in [K]$, $t \in \mathbb{N}$ and $\hat{\underline{L}}_t \in \mathbb{R}_{\geq 0}^K$, it holds that

$$w_{t,i} = \int_1^\infty f(z + \eta_t\hat{\underline{L}}_{t,i}) \prod_{j \neq i} F(z + \eta_t\hat{\underline{L}}_{t,j})\mathrm{d}z = \int_1^\infty \frac{\alpha}{(z + \eta_t\hat{\underline{L}}_{t,i})^{\alpha+1}} \prod_{j \neq i} F(z + \eta_t\hat{\underline{L}}_{t,j})\mathrm{d}z.$$

**Upper bound** The upper bound follows directly from the definition of $w_{t,i}$:

$$\int_1^\infty \frac{\alpha}{(z + \eta_t \underline{\hat{L}}_{t,i})^{\alpha+1}} \prod_{j \neq i} F(z + \eta_t \underline{\hat{L}}_{t,j}) \mathrm{d}z \leq \int_1^\infty \frac{\alpha}{(z + \eta_t \underline{\hat{L}}_{t,i})^{\alpha+1}} \mathrm{d}z \leq \frac{1}{(1 + \eta_t \underline{\hat{L}}_{t,i})^\alpha}.$$

Note that this inequality holds for all $t$, regardless of whether the event $D_t$ occurs.

**Lower bound** Since the cumulative distribution function $F$ takes value in $[0, 1]$, on $D_t$, we obtain for the second term,

$$
\begin{aligned}
w_{t,i} &= \int_1^\infty f(z + \eta_t \underline{\hat{L}}_{t,i}) \prod_{j \neq i} F(z + \eta_t \underline{\hat{L}}_{t,j}) \mathrm{d}z \\
&\geq \int_1^\infty f(z + \eta_t \underline{\hat{L}}_{t,i}) \prod_{j \in [K]} F(z + \eta_t \underline{\hat{L}}_{t,j}) \mathrm{d}z \\
&\geq \int_1^\infty f(z + \eta_t \underline{\hat{L}}_{t,i}) \exp\left( -\sum_{j \in [K]} \frac{1 - F(z + \eta_t \underline{\hat{L}}_{t,j})}{F(z + \eta_t \underline{\hat{L}}_{t,j})} \right) \mathrm{d}z && (\because e^{-\frac{x}{1-x}} \leq 1 - x \text{ for } x < 1) \\
&= \int_1^\infty f(z + \eta_t \underline{\hat{L}}_{t,i}) \exp\left( -\sum_{j \neq i^*} \frac{1 - F(z + \eta_t \underline{\hat{L}}_{t,j})}{F(z + \eta_t \underline{\hat{L}}_{t,j})} \right) \exp\left( -\frac{1 - F(z)}{F(z)} \right) \mathrm{d}z && (\because \underline{\hat{L}}_{t,i^*} = 0 \text{ on } D_t) \\
&\geq \int_{2^{1/\alpha}}^\infty f(z + \eta_t \underline{\hat{L}}_{t,i}) \exp\left( -\sum_{j \neq i^*} \frac{1 - F(z + \eta_t \underline{\hat{L}}_{t,j})}{F(z + \eta_t \underline{\hat{L}}_{t,j})} \right) \exp\left( -\frac{1 - F(z)}{F(z)} \right) \mathrm{d}z \\
&\geq \frac{1}{e} \int_{2^{1/\alpha}}^\infty f(z + \eta_t \underline{\hat{L}}_{t,i}) \exp\left( -2 \sum_{j \neq i^*} (1 - F(z + \eta_t \underline{\hat{L}}_{t,j})) \right) \mathrm{d}z && (\because 2^{1/\alpha} \text{ is the median}) \\
&= \frac{1}{e} \int_{2^{1/\alpha}}^\infty f(z + \eta_t \underline{\hat{L}}_{t,i}) \exp\left( -2 \sum_{j \neq i^*} \frac{1}{(z + \eta_t \underline{\hat{L}}_{t,j})^\alpha} \right) \mathrm{d}z && \text{(Pareto perturbation)} \\
&\geq \frac{1}{e^2} \int_{2^{1/\alpha}}^\infty f(z + \eta_t \underline{\hat{L}}_{t,i}) \mathrm{d}z = \frac{1}{e^2} \frac{1}{(2^{1/\alpha} + \eta_t \underline{\hat{L}}_{t,i})^\alpha}. && \text{(Definition of } D_t \text{ in (10))}
\end{aligned}
$$

Since $\frac{(x+1)^\alpha}{(x+2^{1/\alpha})^\alpha}$ is increasing with respect to $x \geq 0$ for any $\alpha > 1$, this implies that

$$\frac{(1 + \eta_t \underline{\hat{L}}_{t,i})^\alpha}{(2^{1/\alpha} + \eta_t \underline{\hat{L}}_{t,i})^\alpha} \geq \frac{1}{2} \implies \frac{1}{2(1 + \eta_t \underline{\hat{L}}_{t,i})^\alpha} \leq \frac{1}{(2^{1/\alpha} + \eta_t \underline{\hat{L}}_{t,i})^\alpha},$$

which concludes the proof for the lower bound.

**Lower bound for optimal arm** Since $\hat{\underline{L}}_{t,i^*} = 0$ on $D_t$, we obtain that

$$
\begin{aligned}
w_{t,i^*} &= \int_1^\infty \frac{\alpha}{z^{\alpha+1}} \prod_{j \neq i^*} F(z + \eta_t \hat{\underline{L}}_{t,j}) \mathrm{d}z \\
&\geq \int_1^\infty \frac{\alpha}{z^{\alpha+1}} \exp\left( -\sum_{j \neq i^*} \frac{1 - F(z + \eta_t \hat{\underline{L}}_{t,j})}{F(z + \eta_t \hat{\underline{L}}_{t,j})} \right) \mathrm{d}z \qquad (\because e^{-\frac{x}{1-x}} \leq 1 - x \text{ for } x < 1) \\
&\geq \int_{2^{1/\alpha}}^\infty \frac{\alpha}{z^{\alpha+1}} \exp\left( -\sum_{j \neq i^*} \frac{1 - F(z + \eta_t \hat{\underline{L}}_{t,j})}{F(z + \eta_t \hat{\underline{L}}_{t,j})} \right) \mathrm{d}z \\
&\geq \int_{2^{1/\alpha}}^\infty \frac{\alpha}{z^{\alpha+1}} \exp\left( -2 \sum_{j \neq i^*} (1 - F(z + \eta_t \hat{\underline{L}}_{t,j})) \right) \mathrm{d}z \\
&\geq \frac{1}{e} \int_{2^{1/\alpha}}^\infty \frac{\alpha}{z^{\alpha+1}} \mathrm{d}z = \frac{1}{2e},
\end{aligned}
$$

which concludes the proof. $\qquad\square$

**Lemma D.3.** *On $D_t^c$, $\sum_{i \neq i^*} \Delta_i w_{t,i} \geq \frac{1 - e^{-1/2}}{2} \Delta_{\min}$.*

*Proof.* By definition, we have on $D_t^c$ that

$$
\begin{aligned}
w_{t,i^*} &= \int_1^\infty \frac{\alpha}{(z + \eta_t \hat{\underline{L}}_{t,i^*})^{\alpha+1}} \prod_{j \neq i^*} \left( 1 - \frac{1}{(z + \eta_t \hat{\underline{L}}_{t,j})^\alpha} \right) \mathrm{d}z \\
&\leq \int_1^\infty \frac{\alpha}{(z + \eta_t \hat{\underline{L}}_{t,i^*})^{\alpha+1}} \exp\left( -\sum_{j \neq i^*} \frac{1}{(z + \eta_t \hat{\underline{L}}_{t,j})^\alpha} \right) \mathrm{d}z \\
&= \int_1^{2^{1/\alpha}} \frac{\alpha}{(z + \eta_t \hat{\underline{L}}_{t,i^*})^{\alpha+1}} \exp\left( -\sum_{j \neq i^*} \frac{1}{(z + \eta_t \hat{\underline{L}}_{t,j})^\alpha} \right) \mathrm{d}z + \int_{2^{1/\alpha}}^\infty \frac{\alpha}{(z + \eta_t \hat{\underline{L}}_{t,i^*})^{\alpha+1}} \mathrm{d}z \\
&\leq \frac{1}{\sqrt{e}} \int_1^{2^{1/\alpha}} \frac{\alpha}{(z + \eta_t \hat{\underline{L}}_{t,i^*})^{\alpha+1}} + \int_{2^{1/\alpha}}^\infty \frac{\alpha}{(z + \eta_t \hat{\underline{L}}_{t,i^*})^{\alpha+1}} \mathrm{d}z \qquad \text{(by definition of } D_t^c) \\
&\leq \frac{1}{\sqrt{e}} \int_1^{2^{1/\alpha}} \frac{\alpha}{z^{\alpha+1}} + \int_{2^{1/\alpha}}^\infty \frac{\alpha}{z^{\alpha+1}} \mathrm{d}z = \frac{e^{-1/2}}{2} + \frac{1}{2}.
\end{aligned}
$$

Since $1 - w_{t,i^*} = \sum_{i \neq i^*} w_{t,i}$, the result follows. $\qquad\square$

**Lemma D.4** (Lemma of [Rouyer & Seldin (2020)](#)). *Let $T_0 := \max_{i \neq i^*} \left\lceil D\left(\frac{x}{\Delta_i}\right)^2 \right\rceil$ for some constants $x \in \mathbb{R}_{>0}$ and $D \geq 1$. For each suboptimal arm $i \neq i^*$, define $S_i(T) := \Delta_i^{1-\alpha} \sum_{t=T_0+1}^T t^{-\frac{\alpha}{2}}$. Then $S_i(T)$ converges as $T \to \infty$ if and only if $\alpha > 2$. Moreover, for $\alpha > 2$, we have*

$$
\lim_{T \to \infty} S_i(T) \leq \frac{2}{\alpha - 2} \frac{\left( x\sqrt{D} \right)^{2-\alpha}}{\Delta_i}.
$$

# E. Additional experiments

In this section, we present additional experimental results.

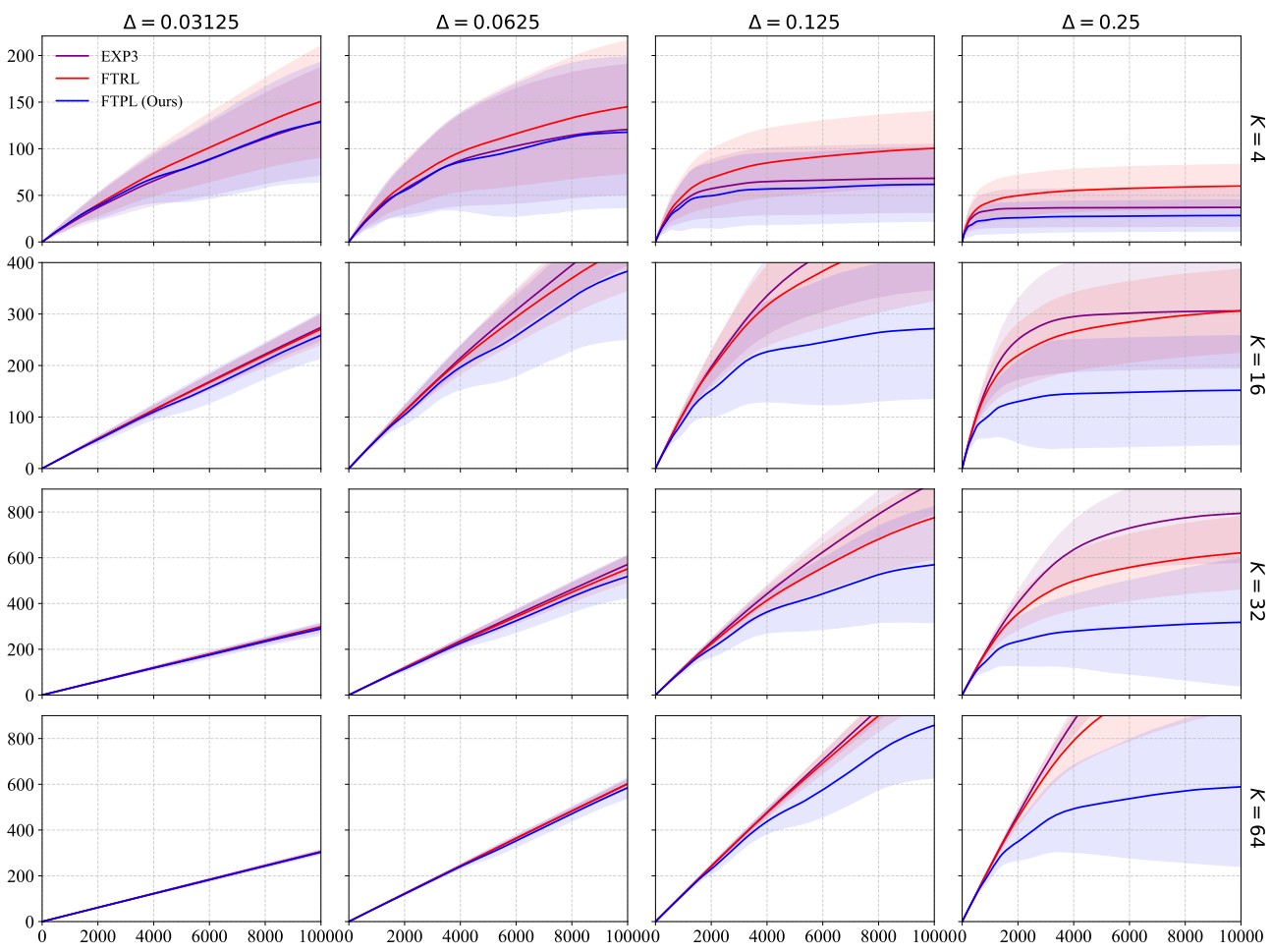

*Figure 6.* Cumulative regret comparison across 16 distinct instances with varying arm counts $K$ and gaps $\Delta$

### E.1. Implementation details

For Decoupled-Tsallis-INF, we compute $w_t$ by using Newton's method with a warm start, as described in Zimmert & Seldin (2021), which can efficiently solve the associated convex optimization problem. For EXP3, one can solve the optimization problem of FTRL with Shannon entropy, where the solution, $w_{t,i}$ can be written as for some $\nu \in \mathbb{R}$

$$w_{t,i} = \exp\left(-\nu - \eta_t \hat{L}_{t,i}\right), \, \forall i \in [K], t \in \mathbb{N} \implies w_{t,i} = \frac{\exp\left(-\eta_t \hat{L}_{t,i}\right)}{\sum_{j=1}^{K} \exp\left(-\eta_t \hat{L}_{t,j}\right)},$$

which can be directly updated after observing $\hat{\ell}_t$. For our proposed policy, we implement a naive baseline that performs a full sorting operation at each round. However, this is unnecessarily inefficient since the loss estimator is updated only for the selected arm $i_t$, and therefore the ranks of the remaining arms change by at most one position. Consequently, instead of re-sorting all arms, it suffices, whenever $\sigma_{t,i_t} \neq 1$, to check whether the rank $\sigma_{t+1,i_t}$ changes using a selection algorithm, which incurs $\mathcal{O}(K)$ time rather than the $\mathcal{O}(K \log K)$ cost of full sorting. As mentioned in Section 4.1, such operations can be efficiently performed using $\Theta(K)$ memory, by storing the previous rank. The full implementation is publicly available at https://github.com/chaiwonkim/FTPL-for-Decoupled-Bandits.

### E.2. Adversarial regime

To demonstrate the scalability and robustness of our policy, we conduct a comprehensive evaluation across a broader range of environments. Specifically, we expand our experimental scope to 16 distinct instances by varying the arm count

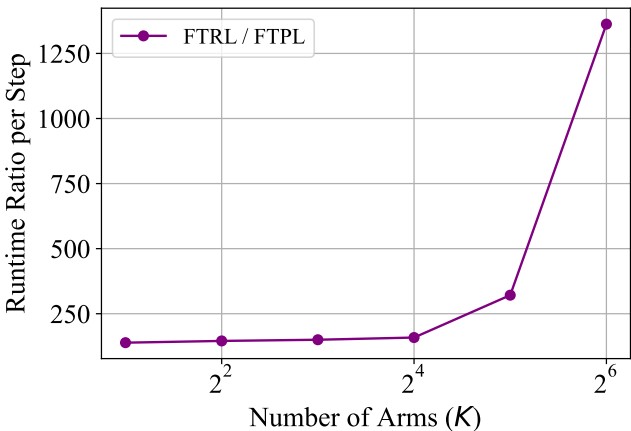

*Figure 7.* Runtime ratio using the SCS

$K \in \{4, 16, 32, 64\}$ and the gap $\Delta \in \{0.03125, 0.0625, 0.125, 0.25\}$, following the experimental framework of Zimmert & Seldin (2021). Figure 6 shows the average cumulative regret over 1,000 independent repetitions, with shaded regions indicating one standard deviation. Across all instances, our policy consistently outperforms all baselines, maintaining superior performance even under increased problem complexity (i.e., with a large $K$ and a small $\Delta$).

Figure 7 shows the per-step runtime ratio of FTRL to FTPL for $K \in \{2^i : i \in [6]\}$, where the optimization step in FTRL is solved using the splitting conic solver (SCS). The $x$-axis is shown on a logarithmic scale. As the number of arms increases, the ratio grows rapidly, reaching approximately 1363 for $K = 64$. Due to the excessive runtime of FTRL for larger number of arms, the experiments are repeated only 100 times and are not performed beyond $K = 64$.

### E.3. Real-world data-based simulation

To further validate our approach in real-world scenarios, we construct a robust offline oracle by transforming the MovieLens 100K dataset, a common benchmark for online learning (Li et al., 2019), into an adversarial decoupled bandit environment. The rewards are estimated via a matrix factorization-based approach. To address the challenge of counterfactual evaluation and prevent the oracle from producing biased rewards, we apply a double-centering preprocessing step prior to factorization, which allows the model to capture pure latent user-item interactions.

We conduct all experiments with 1,000 independent repetitions across varying movie counts ($K \in \{32, 64, 128, 256\}$). Figure 8 shows the average cumulative regret, where the shaded region denotes one standard deviation. As illustrated, our proposed policy consistently achieves superior regret performance, significantly outperforming both FTRL and EXP3 across all tested environments.

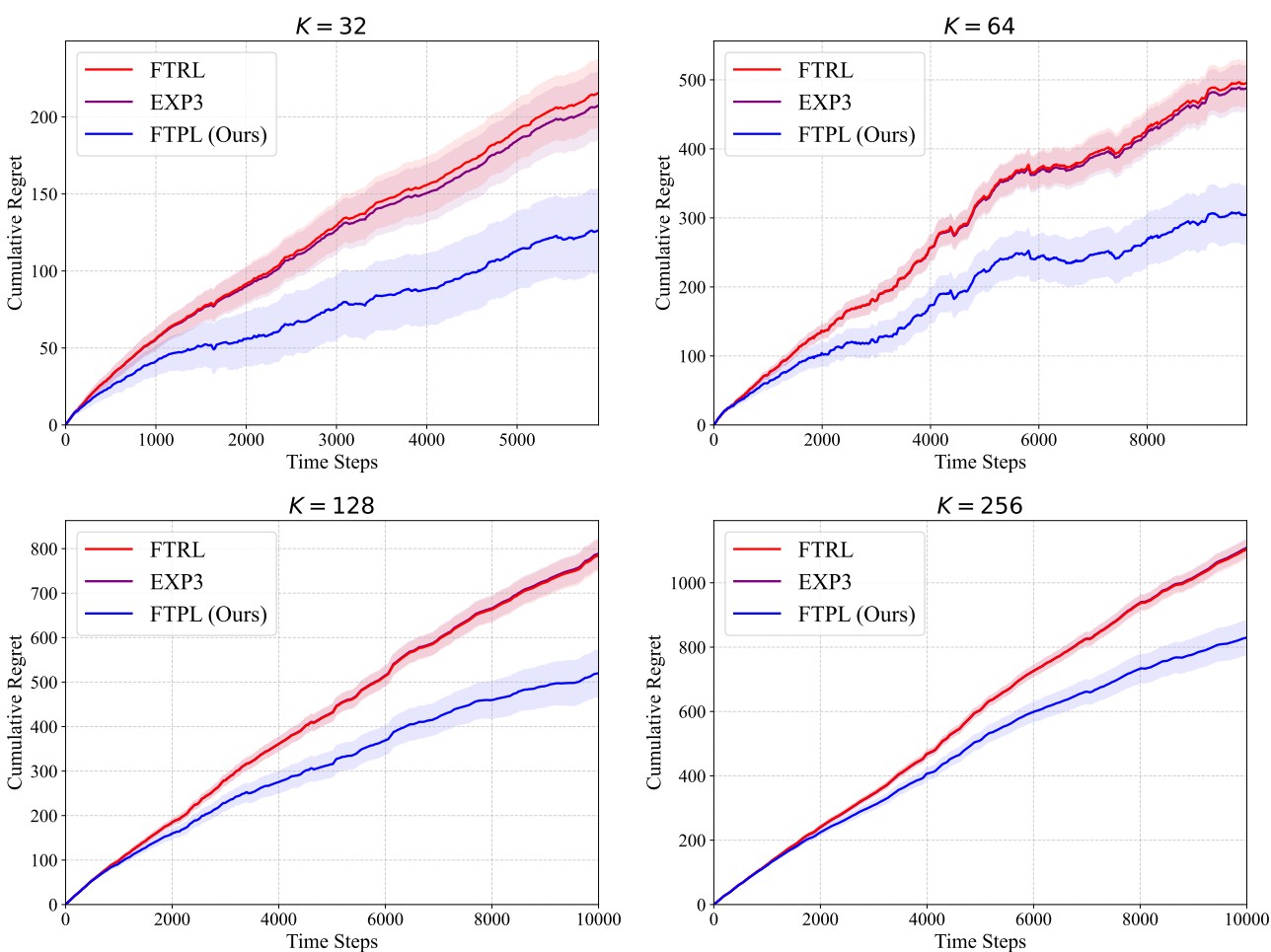

*Figure 8.* Cumulative regret comparison on the MovieLens 100K dataset under varying numbers of movies $K$

