# OpenReview forum: "Follow-the-Perturbed-Leader for Decoupled Bandits: Best-of-Both-Worlds and Practicality"
_ICML.cc/2026/Conference — ICML 2026 regular_

### Official Review · Reviewer_bCFx · 2026-03-12

**Soundness:** 3
**Presentation:** 3
**Significance:** 2
**Originality:** 2
**Overall Recommendation:** 4
**Confidence:** 3

**Summary:**

This paper studies the docoupled multi-armed bandit problem and propose a FTPL-based method that achieves best of both worlds guarantee with improved computation and runtime over existing works. Emprirical results are finally provided to validate the high efficiency of the proposed method.

**Compliance With Llm Reviewing Policy:**

Affirmed.

**Final Justification:**

I improved the rating of this work as mentioned in my reply to the authors' responses from 3 to 4.

**Key Questions For Authors:**

Please read my main concern above

**Strengths And Weaknesses:**

Strengths:

1. This paper studies an interesting and timely problem, and the writing is mostly clear and easy to ready.
2. This paper provided a suite of empirical results, and the proposed methods are proved to be computationally efficient.

Weaknesses:
I am not very familiar with the decoupled bandit and best of best world bandit, so I will refer to other reviewers' comments to adjust my rating accordingly.

I may overlook, but the biggest advantage of the proposed method is indeed the computational benefit but not any theoretical gain, and hence the scope is a bit limited. And I read the design of the new FTPL method, which is also not new and novel to me.

---

> ### Author Rebuttal · Authors · 2026-03-30
>
> We thank the reviewer for the time and effort devoted to evaluating our paper. We also appreciate the reviewer’s candid note regarding (un)familiarity of the relevant literature.
> We would like to clarify the following two key contributions to ensure that the key practical and theoretical advances of this work paper are fully recognized.
>
> For convenience, we provide all additional results at this [anonymous link](https://anonymous.4open.science/r/anonymous1-D49F/Figures.pdf).
>
> ---
>
> ### 1. Novelty of Our Surrogate Arm Selection Probability
>
> Regarding the novelty of our policy, we emphasize that our construction of $q_t$ represents an advancement in two key ways: **(i)** it provides an **efficiently computable surrogate** that replaces the explicit computation of $w_t$ without requiring optimization or resampling, and **(ii)** it may provide **new directions in FTPL research** on adaptive learning rates, an area where progress has been constrained by the absence of closed-form arm-selection probabilities.
>
> **(i)** This contribution is non-trivial, as the direct use of geometric resampling (GR) to compute **full arm-selection probability vector introduces redundant computations** in bandit settings. Since GR is designed to estimate the probability of only the played arm, a naive adaptation that estimates all $K$ coordinates via repeated resampling would incur at least $O(K^2 \log K)$ per-round complexity. In line with this motivation, our novel approach achieves $O(K)$ per-round complexity by removing the need for GR entirely. Consequently, our efficiently computable method for approximating the arm selection probability $w_t$ allows **FTPL to fully realize its computational potential**, positioning it as an efficient alternative to FTRL. Importantly, this efficiency is achieved without sacrificing theoretical guarantee: our approach simultaneously maintains BOBW guarantees and enhances empirical performance compared to FTRL baseline (see #2 below for details), advancements that we believe are non-trivial.
>
> While the initial submission utilized a naive sorting-based implementation ($O(K \log K)$) as noted in Line 244, we have since **implemented an optimized version** that reduces the worst-case complexity to $O(K)$ via incremental rank updates (discussed in Appendix E.1). These updated plots, available in Figure A of above link, will be fully incorporated into the revised version.
>
> **(ii)** This is particularly important as recent FTRL approaches in the BOBW literature, including applications in graph feedback and partial monitoring, rely explicitly on $w_t$ (Tsuchiya & Ito, 2024). By extension, we expect that our approximation technique for $w_t$ provides a theoretical foundation for the principled design of adaptive learning rates in FTPL. This could potentially enable FTPL to obtain strong theoretical guarantees and broader applicability in complex environments.
>
> Moreover, as recent FTRL methods increasingly utilize hybrid regularizers, Newton’s method may not always be readily applicable, making a computationally practical alternative important. Our surrogate $q_t$ addresses this limitation by replacing the dependence on $w_t$ that FTRL requires, thereby offering a clear conceptual and technical contribution. This may provide new directions for deploying FTPL in complex settings that extend well beyond the decoupled setting.
>
> ---
>
> ### 2. Empirical Performance in Various Settings
>
> To validate our policy’s robustness and efficiency, we conducted additional experiments across a broader range of instances and real-world scenarios.
>
> - **Robustness across arm counts and gaps**
>
> While our initial submission focused on specific configurations from Zimmert & Seldin (2021), following Honda et al. (2023), we have since expanded our experimental scope. Specifically, we evaluated our policy across **16 additional instances** by varying the arm count and the gap, where we followed the entire experimental settings in Zimmert & Seldin (2021). As shown in Figure E, **our policy consistently outperforms all baselines**, even as the problem complexity increases (Large $K$ and small gap).
>
> - **Real-world data-based simulation**
>
> We constructed a **robust offline oracle using the MovieLens 100K dataset**, a standard benchmark for online learning [1]. By applying matrix factorization-based reward estimation, we transformed the dataset into an adversarial decoupled environment where user ratings serve as rewards. To address the counterfactual evaluation challenge and prevent biased rewards, we employed Double Centering to capture pure latent user-item interactions. All experiments were conducted with 100 independent runs across varying arm (movie) counts $K$. As shown in Figure D, our proposed policy **consistently achieves better regret performance** than all baselines, with shaded regions denoting one standard deviation.
>
> [1] Li, S., Lattimore, T., and Szepesvari, C. Online learning to rank with features. ICML 2019.

---

> > ### Author Rebuttal · Reviewer_bCFx · 2026-04-03
> >
> > Thank you for your response. I now have a more positive view on this work, and will adjust my rating in the final justification and discussion with other reviewers.

---

> > > ### Author Response · Authors · 2026-04-03
> > >
> > > We are glad that our responses have fully addressed your feedback. We greatly appreciate your positive view on our work and your decision to adjust the score in the final justification.

---

### Official Review · Reviewer_8vyd · 2026-03-12

**Soundness:** 3
**Presentation:** 4
**Significance:** 3
**Originality:** 3
**Overall Recommendation:** 5
**Confidence:** 2

**Summary:**

The authors consider the decoupled bandit problem where at time $t$ a learner selects $i_t,j_t$ from a set of arms $[K]$, observes the loss $\ell_{t,j_t}$ and suffers the (unobserved) loss $\ell_{t,i_t}$, where the goal is to minimize regret with respect to the best arm in hindsight

$$
R_T = \sum_{t=1}^{T} \mathbb{E}( \ell_{t,i_{t}}   ) - \min_{i \in [K]}  \left(  \sum_{t=1}^{T} \mathbb{E}( \ell_{t,i}   ) \right)
$$
For this problem, the authors propose a novel algorithm based on the "follow the perturbed leader" idea that performs close to optimally both in the adversarial and stochastic regimes simultaneously and that has low computational complexity.  The main technical contribution is a scheme that allows to estimate arm selection probabilities without using resampling (from prior work), because in FTPL one chooses a random perturbation rather than an arm selection probabilities, so that one must compute the order statistics of the random perturbation and this is usually difficult.

**Compliance With Llm Reviewing Policy:**

Affirmed.

**Final Justification:**

I already had a positive impression of the paper before the rebuttal and discussion, and I hence have maintained my already high score.

**Key Questions For Authors:**

See remarks above:
- Can the numerical experiments be fixed ?

**Limitations:**

Yes

**Strengths And Weaknesses:**

Overall the paper looks good, the writing is excellent, and the theoretical results seem solid. The main shortcomings seems to be: the numerical experiments, and the fact that the authors do not really adress how "inefficient" the algorithms from prior work are (their numerical experiments seem to suggest that they have the same computational complexity of $O(K)$ per iteration which is optimal, just with a different pre-factor), since one of the main motivations for their work is to develop algorithms that are both computationally efficient and "best of both worlds" optimal.

Detailed remarks

- Page 2: "However, this result is highly suboptimal, as even an anytime sampling rule designed for pure exploration tasks attains a time-independent cumulative regret of $\mathcal{O}(K^3/\Delta_{\min}^2 )$" This phrase is a bit misleading because $\Delta_{\min}$ can scale with $T$, and the bound only makes sense in the regime where $\Delta_{\min} \ge \frac{1}{\sqrt{T}}$
- Page 2:  " To estimate this, FTPL policies usually employ geometric resampling (Neu \& Bartok, 2016) or its efficient variant (Chen et al., 2025), which incur a perstep average cost of $O(K^2)$ or $O(K \log(K))$, respectively." If this is correct I do not understand what is wrong with prior work: as far as I understand, given a distribution over $K$ arms, the time needed to sample from said distribution is at least $\Omega(K)$, unless the distribution has some special structure, and this is assuming that the distribution can be computed instantly. Hence any FTPL algorithm requires at least time $O(K)$ to be implemented, and the results from prior work are provably optimal up to a multiplicative $\log(K)$.
- Page 3: I find the "stochastically constrained adversarial regime" to be rather artificial. At this rate one may just assume that rewards are i.i.d. stochastic.
- Page 3: Is the assumption that the optimal arm is unique necessary ?
- Page 4: The authors argue that the strategy of Rouyer \& Seldin (2020) is complex because it involves an optimization problem. However they do not provide an evaluation of its computational complexity in terms of $K$. The optimization problem is a convex optimization problem over the simplex, so there could very well exist algorithms that solve them efficiently (say in time $O(K)$).
- Page 5: "Theoretical guarantee" typo
- Page 5: When combining Theorem 3.1. and Theorem 3.2. one observes that in both cases $\alpha = 3$ yields the best bounds. In that case, why not simply choose this value by default ? If this is not always the best choice, a discussion could be added here about the drawbacks of $\alpha \ne 3$.
- Page 7: The results of Figure 1 do not demonstrate that the proposed algorithm is better than EXP3 or FTRL, because the confidence interval for the regret of the proposed algorithm intersects with the confidence intervals for the regret of EXP3 and FTRL.
- Page 7: The runtime comparison in Figure 3 is both disapointing and contradictory to the authors claims that previous strategies for implementing FTRL (Newton's method) are not efficient computationally. Indeed, there seems to be at most a constant factor between FTRL and the algorithm proposed by the authors. Why is that so ?
- Page 8: The same problem with confidence intervals occurs in Figures 4 and 5.

---

> ### Author Rebuttal · Authors · 2026-03-30
>
> Thank you for the time and effort you devoted to reviewing our paper. As you pointed out, we will revise the discussion on Page 2 clearly to avoid potential misleading. Below, we provide detailed responses to the weakness you raised.
>
> For convenience, we provide all additional results at this [anonymous link](https://anonymous.4open.science/r/anonymous1-D49F/Figures.pdf).
>
> ---
>
> ## Theoretical Per-Round Complexity
>
> - **FTPL with resampling**
>
> The GR/CGR are designed to estimate the probability of the **played arm**, which is sufficient for IW estimator in standard bandits with O(KlogK) time. In decoupled bandits, however, previous approaches utilize the **full probability vector**. Therefore, a naive adaptation that estimates **all $K$ coordinates** via repeated resampling would incur $O(K^2 \log K)$ average complexity.
>
> - **Comparison with Tsallis-INF**
>
> Our policy achieves an explicit per-round complexity of O(K). While the initial submission used a naive sorting-based implementation to compute the rank with O(KlogK) per-round complexity as noted in Line 244, we have since implemented an **optimized version that reduces the worst-case complexity to O(K)** via incremental rank updates as discussed in App. E.1. This efficiency is achieved by locating the new rank of played arm via binary search in O(logK) and updating the affected arms in O(K) in the worst-case.
>
> In contrast, the total **computational cost of Decoupled Tsallis-INF remains theoretically unclear.** In our setting, optimization can be solved by iterative Newton’s method that costs O(K) per iteration, but as noted by Chen et al. (2025), the required number of iterations to satisfy its regret guarantee has not been formally characterized. Due to this theoretical ambiguity, we provided an empirical runtime comparison.
>
> ---
>
> ## Figure 3
>
> We have improved Figure 3 through two key updates.
>
> First, we re-evaluated the runtime using optimized O(K) implementation discussed above, replacing the naive O(KlogK) sorting-based approach to compute $\sigma$. While the difference between O(K) and O(KlogK) may not be clearly visible in moderate K, the key point is that the two implementations have fundamentally different theoretical complexities. In this context, the FTRL appears to show a similar or slightly less favorable trend than the O(KlogK) implementation due to the iterative nature of Newton’s method.
>
> Second, we addressed the high-variance behavior of previous result, which was largely due to external noise in the computing environment. To mitigate this, we re-ran the experiments using a parallel computing setup that isolates individual runs under identical computational resource. Updated results in Figure A show lower variability compared to Figure 3.
>
> ---
>
> ## Remarks on Page 3
>
> - **SCA regime**
>
> We agree that SCA regime may appear somewhat artificial though it includes the stochastic regime as a special case. In recent BOBW literature, this regime is introduced as an intermediate model that bridges purely stochastic and fully adversarial environments.
>
> - **Unique optimal arm**
>
> The assumption of a unique optimal arm is commonly adopted in BOBW literature for analytical reason (Rouyer & Seldin, 2020; Honda et al., 2023). The main difficulty is on the stability term, where we handle the sum of the stability terms over arms $i \ne i^*$ excluding a single arm. When multiple optimal arms exist, we may require additional techniques (see, Sec 7.1 of Zimmert & Seldin (2021)).
>
> While there exist approaches that remove this assumption in the FTRL framework (e.g., skewed Bregman divergence [1]), many FTRL works in the BOBW literature still rely on the uniqueness assumption (Ito et al., 2024). Extending analysis to handle multiple optimal arms is an interesting direction for future work, but it is currently unclear how to adapt such techniques to the FTPL setting.
>
> [1] Ito, S. Parameter-free multi-armed bandit algorithms with hybrid data-dependent regret bounds. COLT 2021.
>
> ---
>
> ## Choice of $\alpha$
>
> While $\alpha=3$ yields the best bounds, we do not fix this value by default for two reasons. First, any $\alpha \in [2,3]$ achieves a $\Delta$-dependent bounds in SCA regime, with $\alpha=3$ only improving constants. Second, since prior work (Rouyer & Seldin, 2020) considers a range of parameters $\beta \in (0,2/3]$, we include $\alpha \in (1,3]$ to maintain consistency and avoid impressions on restrictive parameter choices. Notably, FTRL with $\beta$-Tsallis entropy is related to FTPL with $1/(1-\beta)$ Fréchet-type perturbations (Kim & Tewari, 2019), which further motivates considering a range of $\alpha$ values. We also provide empirical results with different $\alpha$ in Figure B. As expected, $\alpha=1$ (outside our guarantees) performs poorly, while $\alpha=3$ shows the lowest mean regret.
>
> ---
>
> ## Confidence Interval (CI)
>
> We clarify that the previous shaded regions implies one standard deviation rather than CI. We have included 95% CI (t-dist.) in Figure F.

---

> > ### Author Rebuttal · Reviewer_8vyd · 2026-04-01
> >
> > I thank the authors for their answers which have addressed my previous (minor) criticisms. I will keep my score of 5 (Accept).

---

> > > ### Author Response · Authors · 2026-04-03
> > >
> > > We sincerely thank you for your continued support. We are pleased that our responses have fully addressed your feedback.

---

### Official Review · Reviewer_oEsN · 2026-03-13

**Soundness:** 2
**Presentation:** 3
**Significance:** 2
**Originality:** 2
**Overall Recommendation:** 4
**Confidence:** 3

**Summary:**

This work tackles the problem of decoupled multi-armed bandits. At each round, the learner chooses one arm to explore, observe its loss but not incur it, and another to exploit that incurs loss but the agent do not observe it. That means the learner must coordinate two decisions per round while maintaining unbiased loss estimates. The authors propose a new Follow-the-Perturbed-Leader (FTPL) algorithm leveraging Pareto perturbations. They choose to exploit the arm that minimizes the perturbed cumulative losses, while exploration is sampled from a distribution that approximates a power of the exploitation probability. The authors argue that existing works on decoupled bandits hinges on FTRL policies that must solve an convex optimization per step to get pulling probability over arms, though with FTPL policies they bypass it by using a heavy-tailed perturbation such as Pareto.

**Compliance With Llm Reviewing Policy:**

Affirmed.

**Final Justification:**

In the discussion/rebuttal with authors, they provide significant amount of experiments against my initial review. Fully trusting the integrity of these results, they look promising. Thus, I improve my score to 4 (weak accept) from 3. I read other (reviewer) discussions too. They seem to be resolved as well. I see no immediate major issues at this point with this work, thus I suggest a positive evaluation.

**Key Questions For Authors:**

- It seems to me, the Pareto perturbation is at the core of the algorithm design. Heavy-tailed perturbations are known to enable BOBW guarantee, but there is no sufficient discussion on the choice of Pareto as the perturbation distribution. In other words, can the authors shed some light on the rationale behind choosing such a perturbation?

- In the introduction, the paper is motivated with the aim to build a more *computationally efficient* algorithm for decoupled bandits. But I don't observe any formal discussion of computational complexity guaranties of the proposed algorithm vs. the available strong baselines like Rouyer & Seldin (2020). Since the regret bounds are similar, I would not have mind if they were not formally stated, but a discussion on efficiency becomes imperative.

- The empirical evidence only considers synthetic data. This is very limited in my opinion. If this work claims the proposed algorithm to be *efficient* compared to baselines, they should consider varying size of arm set, also experiments involving real tasks based on decoupled bandits.

- The setting of decoupled bandits is a specialized setting. The paper would benefit from a dedicated discussion in the beginning on where this setting becomes helpful.

- How sensitive the regret upper bound guaranties and empirical performance on the shape parameter of perturbation distribution chosen? Does the algorithm still works if I use other heavy-tailed distribution such as Student-t or Frechet?

**Limitations:**

Yes.

**Strengths And Weaknesses:**

**Strengths:** The paper is relatively well-written, related work is inclusive, easy to parse. The work is focused on designing a more efficient algorithm for attaining tight or optimal BOBW regret guaranties. Thus, the paper is well-motivated.

**Weaknesses:** See Questions below.

---

> ### Author Rebuttal · Authors · 2026-03-30
>
> Thank you for the time and effort you devoted to reviewing our paper. As you suggested, we will incorporate more detailed discussion on applications beyond those in Introduction. Below, we provide detailed responses to the weakness you raised.
>
> For convenience, we provide all additional results at this [anonymous link](https://anonymous.4open.science/r/anonymous1-D49F/Figures.pdf).
>
> ---
>
> ## Why Pareto?
>
> We adopt Pareto perturbations since they simplify both the analysis and the construction of $p_t$.
>
> While Fréchet-type perturbations are known to achieve BOBW guarantees (e.g., Lee et al., 2025), directly using them introduces technical difficulties. In particular, they lead to bounds of form  $1/\underline{L}$, which is unbounded when $\underline{L}=0$, making the analysis (especially Lemma D.2 relating $q\_t$ and $w\_t$) complicated.
>
> In contrast, Pareto gives bounds of form $1/(\eta \underline{L}+1)$, which remains finite and simplify the analysis. More generally, (asymmetric) distributions of form $S(x)/(1+x)^\alpha$ with slowly varying $S$ can be applied, where Pareto corresponds to the simplest case $S\equiv 1$.
>
> ---
>
> ## Theoretical Per-Round Complexity
>
> We would like to clarify why a closed-form comparison of theoretical complexity is **challenging** for the existing FTRL baseline.
>
> First, our proposed policy achieves a theoretically explicit per-round complexity of O(K). While the initial submission utilized a naive sorting-based implementation to compute the rank with O(K logK) per-round complexity as noted in Line 244, we have since implemented an **optimized version that reduces the worst-case complexity to O(K)** via incremental rank updates as discussed in Appendix E.1. This efficiency is achieved by locating the new rank of played arm via binary search in O(log K) and updating the affected block of arms in O(K) in the worst case.
>
> In contrast, the total **computational cost of Decoupled Tsallis-INF remains theoretically unclear.** This policy relies on an iterative Newton’s method that costs O(K) per iteration, but as noted by Chen et al. (2025), the required number of iterations to satisfy its regret guarantee has not been formally characterized. Due to this theoretical ambiguity, we provide an empirical runtime analysis.
>
> **Figure 3 Update:**  We have re-evaluated the runtime performance using our optimized O(K) implementation, replacing the previous O(K logK) naive sorting-based approach. These updated plots are available in Figure A of above link.
>
> ---
>
> ## Additional Experiments
>
> We initially focused on synthetic experiments, as they are standard in bandit research, where real-world (offline) datasets often do not provide the feedback for all (counterfactual) actions. But, at your requests, we performed additional scenarios based on real-world data.
>
> - **Real-world data-based simulation**
>
> We constructed a **robust offline oracle using the MovieLens 100K dataset**, a standard benchmark for online learning [1]. By applying matrix factorization-based reward estimation, we transformed the dataset into an adversarial decoupled environment where user ratings serve as rewards. To address the counterfactual evaluation challenge and prevent biased rewards, we employed Double Centering to capture pure latent user-item interactions. All experiments were conducted with 100 independent runs across varying arm (movie) counts $K$. As shown in Figure D, our proposed policy consistently achieves better regret performance than all baselines, with shaded regions denoting one standard deviation.
>
> - **Robustness across arm counts and gaps**
>
> While our initial submission focused on specific configurations from Zimmert & Seldin (2021), following Honda et al. (2023), we have since expanded our experimental scope. Specifically, we evaluated our policy across **16 additional instances** by varying the arm count and the gap, where we followed the entire experimental settings in Zimmert & Seldin (2021). As shown in Figure E, our policy consistently outperforms all baselines, even as the problem complexity increases (Large $K$ and small gap).
>
> [1] Li, S., Lattimore, T., and Szepesvari, C. Online learning to rank with features. ICML 2019.
>
> ---
>
> ## Different $\alpha$ and Perturbations
>
> The choice of $\alpha$ affects the constant factors in regret bounds. Larger $\alpha$ increases regret in adversarial regimes, and $\alpha>3$ increases constants in stochastic regimes (Thm 3.2). Empirical results for different $\alpha$ are given in Figure B. As expected, $\alpha=1$ performs poorly due to the lack of a finite first moment, which our analysis does not cover.
>
> While it may be possible to apply other perturbations, it would require redesigning exploration probabilities. Moreover, symmetric perturbations such as student-t do not satisfy required properties for current BOBW analysis (Lee et al. 2025). Results with different perturbations are provided in Figure C, where the student-t shows the worst performance as expected.

---

> > ### Author Rebuttal · Reviewer_oEsN · 2026-04-03
> >
> > I thank the authors for their detailed response. The experimental results looks promising. I will raise my score during my final justification.

---

> > > ### Author Response · Authors · 2026-04-03
> > >
> > > We greatly appreciate your decision to raise the score in the final justification.
> > > Thank you for recognizing the strength of our experimental results.

---

### Official Review · Reviewer_TEqS · 2026-03-13

**Soundness:** 3
**Presentation:** 2
**Significance:** 2
**Originality:** 3
**Overall Recommendation:** 4
**Confidence:** 3

**Summary:**

The paper considers a variant of Multi-Armed Bandits called Decoupled Bandits (where the learner plays two actions---one incurs regret, doesn't reveal outcome, and the other incurs no regret, but reveals outcome---at every time-step.)
Within Decoupled Bandits, it gives an algorithm to solve the problem of Best of Both Worlds (BOBW) where a single algorithm has good regret guarantees in both the adversarial rewards setting and the stochastically constrained adversarial (which is more general than the usual stochastic rewards) setting.
The main contribution of this work is this algorithm which, while having good regret bounds, is also computationally inexpensive.

**Compliance With Llm Reviewing Policy:**

Affirmed.

**Final Justification:**

Pros: The writing is clear, enough context and comparisons with literature is provided.
Cons: The setting does appear quite narrow/niche and could be of interest to a limited audience. MABs --> Decoupled MABs --> Best of Both Worlds --> Computational Efficiency.
The rebuttal cleared most of the other concerns, including an elaborate new experimentation.
I shall keep my score.

**Key Questions For Authors:**

1. The paper argues their method is quicker as it avoids the resampling/convex-optimization steps. Could the authors comment about the running time complexity of their algorithm?
I see the empirical comparisons, but I couldn't find the theoretical run times.


2. The problem setting is niche, feels like a specialization that is 3 or 4 levels deep---MABs --> Decoupled MABs --> Best of Both Worlds --> Computational Efficiency.
The paper mentions that the design of exploration probabilities, $p_t$, can be on independent interest. Could the authors comment about the other problems where this technique can be applicable?

3. I perhaps got this wrong. In Corollary 3.3, should it be $\sum_{i \neq i^*} \frac{1}{\Delta_i^2}$? Squared instead of just $\Delta_i$. A plain substitution from Theorem 3.2 appears to yield so.

4. In Figure 3, why is the run time erratic for some specific $K$s? Could you also plot the standard deviation?

5. It is *not* a criticism that the experiments are run on only synthetic instances (and not real world datasets).
However, could the authors give some rationale about the specific choice of instances and arm counts? It appears arbitrary from a reader's perspective.

**Limitations:**

Not applicable, the work is primarily theoretical.

**Strengths And Weaknesses:**

Strengths:
1. Computational aspects of learning algorithms are an important problem to study. This paper's contributions are significant in that sense.

2. Subsection 2.2 was useful in helping the reader understand the context of this work. Enough explanations are given about how $w$ and $p$ probabilities are computed in other works.

Weakness:
1. The paper is difficult to be independently read. While the paper does not have a dedicated Related Works section, a lot of related work is mentioned and cited throughout the body of the paper. It is an opinion that the paper might benefit by having these in a dedicated Related Works section and having the main body leaner.

Some of the 'Key Questions' below also highlight some weaknesses.

Aside,
I did not thoroughly check the correctness of the proofs.

---

> ### Author Rebuttal · Authors · 2026-03-30
>
> Thank you for the time and effort you devoted to reviewing our paper. As you suggested, we will incorporate detailed related work section in the revised version for better readability. Below, we provide detailed responses to the questions you raised.
>
> For convenience, we provide all additional results at this [anonymous link](https://anonymous.4open.science/r/anonymous1-D49F/Figures.pdf).
>
> ---
>
> ## Theoretical Per-Round Complexity
>
> We would like to clarify why a closed-form comparison of theoretical complexity is **challenging** for the existing FTRL baseline.
>
> First, our proposed policy achieves a theoretically explicit per-round complexity of O(K). While the initial submission utilized a naive sorting-based implementation to compute the rank with O(K logK) per-round complexity as noted in Line 244, we have since implemented an **optimized version that reduces the worst-case complexity to O(K)** via incremental rank updates as discussed in Appendix E.1. This efficiency is achieved by locating the new rank of played arm via binary search in O(log K) and updating the affected block of arms in O(K) in the worst case.
>
> In contrast, the total **computational cost of Decoupled Tsallis-INF remains theoretically unclear.** This policy relies on an iterative Newton’s method that costs O(K) per iteration, but as noted by Chen et al. (2025), the required number of iterations to satisfy its regret guarantee has not been formally characterized. Due to this theoretical ambiguity, we provide an empirical runtime analysis in Figure 3.
>
> **Figure 3 Update**:  We have re-evaluated the runtime performance using our optimized O(K) implementation, replacing the previous O(K logK) naive sorting-based approach. These updated plots, available in Figure A of above link, will be fully incorporated into the revised version.
>
> ---
>
> ## Extension of Exploration Probabilities
>
> Recently, many adaptive bandit algorithms, particularly the FTRL framework for BOBW guarantee, rely on explicit access to the probability vector $w_t$. For instance, several recent advanced BOBW results often utilize $w_t$ to design adaptive learning rates, including works on partial monitoring (Tsuchiya et al., 2023), graph bandits (Ito et al., 2024). Similarly, in heavy-tailed bandits, $w_t$ is used to define truncation thresholds for controlling extreme losses [1].
>
> In this context, our approach shows that it is possible to replace the explicit computation of $w_t$ with a directly computable surrogate $q_t$, while still maintaining regret guarantees. Therefore, our contribution is not only a more efficient BOBW algorithm in the FTPL framework, but also a **step toward recovering the adaptivity of FTPL frameworks without requiring explicit computation for $w_t$**.
>
> [1] Huang, J., Dai, Y., and Huang, L. Adaptive best-of-both-worlds algorithm for heavy-tailed multi-armed bandits. ICML 2022.
>
> ---
>
> ## Corollary 3.3
>
> A direct substitution introduces a $\Delta^{-2}$ dependence, as the reviewer pointed out. However, by splitting the regret summation at a threshold $t’$ and bounding the initial terms (up to $t’$) with the $O(\sqrt{Kt’})$ adversarial regret, we can reduce the overall sum by appropriately choosing $t’$.
>
> ---
>
> ## Figure 3
>
> The erratic behavior in Figure 3 was primarily due to the experiments being conducted on a personal laptop, where unpredictable background processes introduced additional variability in runtime for certain values of K.
>
> To mitigate this effect, we re-ran the experiments using a controlled parallel computing setup that isolates individual runs to reduce such external noise. These updated experiments utilize the optimized O(K) implementation discussed above. As requested, we now include the standard deviation in Figure A-(ii).
>
> ---
>
> ## On Experiments
>
> - **Rationale for synthetic instances**
>
> First, to ensure fair comparison and avoid potential concerns about cherry-picking, we **adopted previously used experimental setups** rather than designing new instances (except Figure 5). Specifically, we adopted the adversarial setup from Zimmert & Seldin (2021) and the stochastic setup from the BAI literature (Jourdan et al., 2023).
>
> Second, we focused on synthetic instances, as they are standard in bandit research, where real-world datasets often do not provide the feedback even for the selected actions.
>
> - **Additional Experiments**
>
>     1. **16 additional instances**: To further demonstrate robustness, we evaluated our policy across a broader range of instances using parameters from Zimmert & Seldin (2021). As shown in Figure E, our method consistently outperforms all baselines.
>
>     2. **Real-world data-based simulation**: To further validate our approach, we evaluated our policy in Figure D using a robust offline oracle constructed from the MovieLens 100K dataset, a standard benchmark for online learning [1].
>
> [1] Li, S., Lattimore, T., and Szepesvari, C. Online learning to rank with features. ICML 2019.

---

> > ### Author Rebuttal · Reviewer_TEqS · 2026-04-02
> >
> > The reviewer thanks the authors for their clarifications. There is a lot of new results provided, I couldn't check all of them.
> > I shall keep my score.

---

> > > ### Author Response · Authors · 2026-04-03
> > >
> > > We sincerely thank you for your continued support. We are pleased that our responses have fully addressed your feedback.

---

### Decision · Program_Chairs · 2026-04-30

**Decision:**

Accept (regular)

**Comment:**

The paper presents an efficient FTPL algorithm for best-of-both-worlds decoupled bandits.

Strength:
- The paper is well-written.
- It is valuable to have a working FTPL approach to best-of-both-worlds problems as an alternative to FTRL.
- The original submission has provided very limited empirical evaluation, but in the rebuttal the authors provided a much more satisfying evaluation.

Weaknesses:
- Empirically the Newton's optimization in Tsallis-INF converges within a few iterations, and the cost of each iteration is O(K). The experiments in the paper and the rebuttal also seem to suggest that the runtime improvement of FTPL against FTRL is more like a constant factor. So the claim about computational gain should probably be adjusted accordingly.

Overall, based on positive reaction of the reviewers, I recommend acceptance.